# Development of a targeted BioPROTAC degrader selective for misfolded SOD1

Christen G. Chisholm [1] ✉, Rachael Bartlett [1], Mikayla L. Brown [1],
Emma-Jayne Proctor[1], Natalie E. Farrawell[1], Jody Gorman[1], Fabien Delerue [2,3],
Lars M. Ittner[2,4], Kara L. Vine-Perrow [1], Heath Ecroyd [1], Neil R. Cashman [5,6],
Darren N. Saunders [7], Luke McAlary[1], Jeremy S. Lum [8] ✉ & Justin J. Yerbury[1,9]

The accumulation of misfolded proteins underlies a broad range of neurodegenerative diseases, including amyotrophic lateral sclerosis (ALS). Due to their dynamic nature, these misfolded proteins have proven challenging to target therapeutically. Here, we specifically target misfolded disease variants of the ALS-associated protein superoxide dismutase 1 (SOD1), using a biological proteolysis targeting chimera (BioPROTAC) composed of a SOD1-specific intrabody and an E3 ubiquitin ligase. Screening of intrabodies and E3 ligases for optimal BioPROTAC construction reveals a candidate capable of degrading multiple disease variants of SOD1, preventing their aggregation in cells. Using CRISPR/Cas9 technology to develop a BioPROTAC transgenic mouse line, we demonstrate that the presence of the BioPROTAC delays disease progression in the SOD1^G93A mouse model of ALS. Delayed disease progression is associated with protection of motor neurons, a reduction of insoluble SOD1 accumulation and preservation of innervated neuromuscular junctions. These findings provide proof-of-concept evidence and a platform for developing BioPROTACs as a therapeutic strategy for the targeted degradation of neurotoxic misfolded species in the context of neurodegenerative diseases.

Targeted protein degradation (TPD) is an emerging therapeutic strategy that leverages endogenous protein quality control systems to selectively degrade disease-associated proteins. TPD utilises bifunctional molecules in an induced proximity approach[1], which brings together target proteins and key players in the ubiquitin proteasome (UPS) or the lysosomal autophagy degradation pathways, to degrade target proteins[2]. The induced proximity strategy has the potential to revolutionise the treatment of neurodegenerative diseases characterised by the accumulation of misfolded or toxic proteins, including Alzheimer's disease, Parkinson's disease and amyotrophic lateral

sclerosis (ALS), by promoting the clearance of aberrant misfolded proteins.

There are currently over 200 ALS-associated mutations identified in the *SOD1* gene[3] that interfere with the folding of the encoded protein, destabilising the structure, and resulting in aggregation and the acquisition of a toxic function[4,5]. Several studies have also implicated misfolded forms of wild-type (WT) SOD1[6-10] in protein aggregation and gain-of-toxic function in ALS, although this remains controversial[11,12]. What is clear is that the misfolding and accumulation of toxic SOD1 species disrupts various cellular functions and is central to disease

[1]Molecular Horizons and School of Science, University of Wollongong, Wollongong, NSW, Australia. [2]Dementia Research Centre, Macquarie Medical School, Faculty of Medicine, Health and Human Sciences, Macquarie University, Sydney, NSW, Australia. [3]Department of Genetics, The University of Texas MD Anderson Cancer Centre, Houston, TX, USA. [4]Celosia Therapeutics, Sydney, NSW, Australia. [5]Djavad Mowafaghian Centre for Brain Health, University of British Columbia, Vancouver, BC, Canada. [6]ProMIS Neurosciences, Toronto, ON, Canada. [7]School of Medical Sciences, Faculty of Medicine and Health, University of Sydney, Sydney, NSW, Australia. [8]Molecular Horizons and School of Medical, Indigenous and Health Sciences, University of Wollongong, Wollongong, NSW, Australia. [9]Deceased: Justin J. Yerbury. ✉e-mail: christen@uow.edu.au; jlum@uow.edu.au

pathogenesis[13–15], making misfolded SOD1 a compelling target for therapeutic strategies aimed at facilitating protein degradation.

A key challenge in targeting toxic SOD1 species for degradation is the need to preserve the natively folded WT species. SOD1 knockout mice exhibit early onset sarcopenia and increased incidence of hepatocarcinoma[16], and reports of rapid disease progression and early death of patients with homozygous mutations for *SOD1* have raised concern over the consequences of targeting WT SOD1[17–19]. Similarly, targeting the WT forms of other neurodegenerative disease-associated proteins, including TDP-43[20] and Huntingtin[21], has accelerated disease progression in both pre-clinical models and in human trials.

Given the need to differentially target misfolded, toxic forms of intracellular SOD1 for degradation, while sparing the natively folded form, TPD provides an attractive therapeutic strategy for ALS. Over the past decade, proteolysis targeting chimeras (PROTACs) have emerged as a promising strategy for the degradation of proteins involved in cancer onset and progression[22–26]. PROTACs are heterobifunctional molecules consisting of two ligands joined by a flexible linker—one ligand binds the target protein and the other recruits an E3 ubiquitin ligase. Bringing these two molecules together in close proximity facilitates the ubiquitination of the target protein and its subsequent degradation by the ubiquitin proteasome system (UPS)[2]. While conventional PROTACs utilise small-molecule ligands to bind target proteins, naturally existing protein binding partners or antibodies specific for the target protein, termed BioPROTACs, have recently been successfully developed[27–29].

In this work, we design and test a series of BioPROTACs specifically targeting misfolded forms of SOD1 for degradation. We explore various chimeras of different SOD1 binding proteins and E3 ligases using a panel of seven single-chain variable fragments (scFvs) derived from monoclonal antibodies that bind to aggregated SOD1 in SOD1-ALS patient tissue[8], soluble aggregated SOD1[30], and toxic seeding species of mutant and WT misfolded SOD1[8,31,32]. Each of these misfolded SOD1 scFvs is fused with a panel of eight proteins possessing the ubiquitination functionality of E3 ligases. Using in vitro assays across three cell lines to screen these panels, we identify a lead BioPROTAC candidate we term MisfoldUbL, as it is specific for misfolded SOD1 and functions via the ubiquitin ligase pathway. In a compound transgenic MisfoldUbL/SOD1[G93A] mouse line, we show that expression of MisfoldUbL delays disease progression, reduces the amount of insoluble SOD1 in the brain, and protects against spinal cord motor neuron loss and denervation at neuromuscular junctions. Together, our results demonstrate that a BioPROTAC can specifically reduce misfolded protein species, leading to disease-modifying effects, supporting this approach in therapeutic applications.

## Results

### BioPROTAC design

ProMIS™ Neurosciences Inc. (Toronto, Canada) had previously generated a panel of seven monoclonal antibodies raised against the electrostatic loop of SOD1 (Fig. 1A), a region of SOD1 that is inaccessible when the protein is natively folded[33]. These antibodies have been validated as specifically binding misfolded forms of SOD1[14,34]. To condense their size for use in a BioPROTAC, we generated single-chain variable fragment (scFv) sequences from these seven antibody clones, comprising $V_H$ and $V_L$ regions joined by a $(G_4S)_3$ linker (Fig. 1A). For the initial scFv screen, the C-terminal catalytic domain of Hsc70-interacting protein (CHIP) was truncated to remove the natural binding domain (CHIPΔTPR), and was then fused to the misfolded SOD1 scFv sequences via a GSGSG linker (Fig. 1B). This resulted in the generation of seven unique chimeric BioPROTACs, referred to herein as BPs 1–7. An scFv to an unrelated protein (β-galactosidase) fused to CHIPΔTPR was used as a control in all in vitro assays. The BioPROTACs were designed to bind misfolded SOD1 via their scFv domain, resulting in ubiquitination by the E3 ligase domain and subsequent proteasomal

degradation of misfolded SOD1. The effectiveness of degradation could then be measured by the depletion in EGFP fluorescence (Fig. 1C).

### BioPROTAC scFvs specifically engage misfolded SOD1 in diverse cell types

We used a panel of in vitro assays to assess the efficacy of BPs 1–7 in reducing cellular levels of aggregation-prone SOD1. We have previously shown that the expression of C-terminally EGFP-tagged SOD1 variants can lead to inclusion formation in cultured cells[35,36]. To this end, HEK293 cells were co-transfected to express either SOD1[WT]-EGFP or the highly aggregation-prone SOD1[A4V]-EGFP with each of the BioPROTACs, and the level of SOD1-EGFP was quantified. Three of the BioPROTACs (BP1, BP4 and BP5) slightly reduced SOD1[WT]-EGFP levels, ranging from 7 to 8% reduction (Fig. 2A). All BioPROTACs except BP7 reduced the amount of misfolded SOD1[A4V]-EGFP in cells compared to the control, ranging from 17 to 38% reduction (Fig. 2B). This reduction was confirmed with immunoblotting (Fig. 2C). The negligible changes in SOD1[WT]-EGFP level compared to the significant changes observed for SOD1[A4V]-EGFP levels strongly indicate specificity of the BioPROTACs towards misfolded forms of SOD1, a finding supported by the unchanged levels of endogenous SOD1 in the presence of the BioPROTACs (Supplementary Fig. 1).

Previous work has shown that expression of SOD1 variants in cultured cells leads to the formation of large insoluble aggregates that are reminiscent of the inclusions observed in motor neurons in tissues from ALS patients[37]. Therefore, we investigated the ability of our BioPROTACs to reduce the formation of insoluble aggregates in cells (Fig. 2D). There was a significant reduction in the number of cells with insoluble SOD1[A4V]-EGFP aggregates upon co-expression of BP2 (75 ± 4%), BP3 (76 ± 3%), BP4 (55 ± 5%) and BP6 (73 ± 4%) compared to the control (Fig. 2E), demonstrating the efficacy of these BioPROTACs at reducing toxic SOD1 accumulation. Similar effects were also observed in two neuron-like cell lines; Neuro-2A and SH-SY5Y, with significant reductions in expression levels of SOD1[A4V]-EGFP and SOD1[G93A]-EGFP, and a reduction of insoluble aggregates by one or more BioPROTACs in each assay, including BP2 in all assays (Supplementary Fig. 2).

To investigate the relationship between BioPROTAC expression levels and the reductions previously observed in misfolded SOD1, we analysed cells by immunocytochemistry (Supplementary Fig. 1F). There were significant increases in expression levels of BP1, BP3, BP4, BP5 and BP7 compared to control (Fig. 2F). After normalising for the level of expression, all the BioPROTACs were found to reduce SOD1[A4V]-EGFP levels compared to the control, with BP2 and BP3 having the greatest effect, reducing levels by 78 ± 0.1% and 93 ± 0.1% respectively (Fig. 2G). Comparing the efficacy of BioPROTACs across the assays performed demonstrated that BP2 was the most effective at reducing both misfolded SOD1 levels and aggregation in cells (Fig. 2H).

We next investigated if BP2 could reduce the cellular levels and aggregation of other SOD1 mutants. With over 200 pathogenic mutations identified[19], a common unfolded state exposing the scFv epitope would be beneficial for a broad-based SOD1-directed BioPROTAC to have therapeutic relevance. Using the cell-based aggregation assay described above, we found that expression of BP2 significantly reduced the proportion of cells with insoluble aggregates compared to control across the range of mutant SOD1 variants tested (A4V, G93A, G85R, D90A, V148G, H46R, G37R, C6G and E100G) (Fig. 2I). Thus, BP2 has broad specificity for misfolded SOD1 variants. To investigate cell toxicity, we compared cell counts between untransfected cells and cells transfected with BP2 over 24 and 48 h, and found no difference (Fig. 2J). To confirm that BP2 binds preferentially to misfolded forms of SOD1, a co-immunoprecipitation assay was employed. As expected, BP2 bound SOD1[A4V]-EGFP more effectively (7-fold) than SOD1[WT]-EGFP (Fig. 2K), supporting its specificity for the misfolded form of the

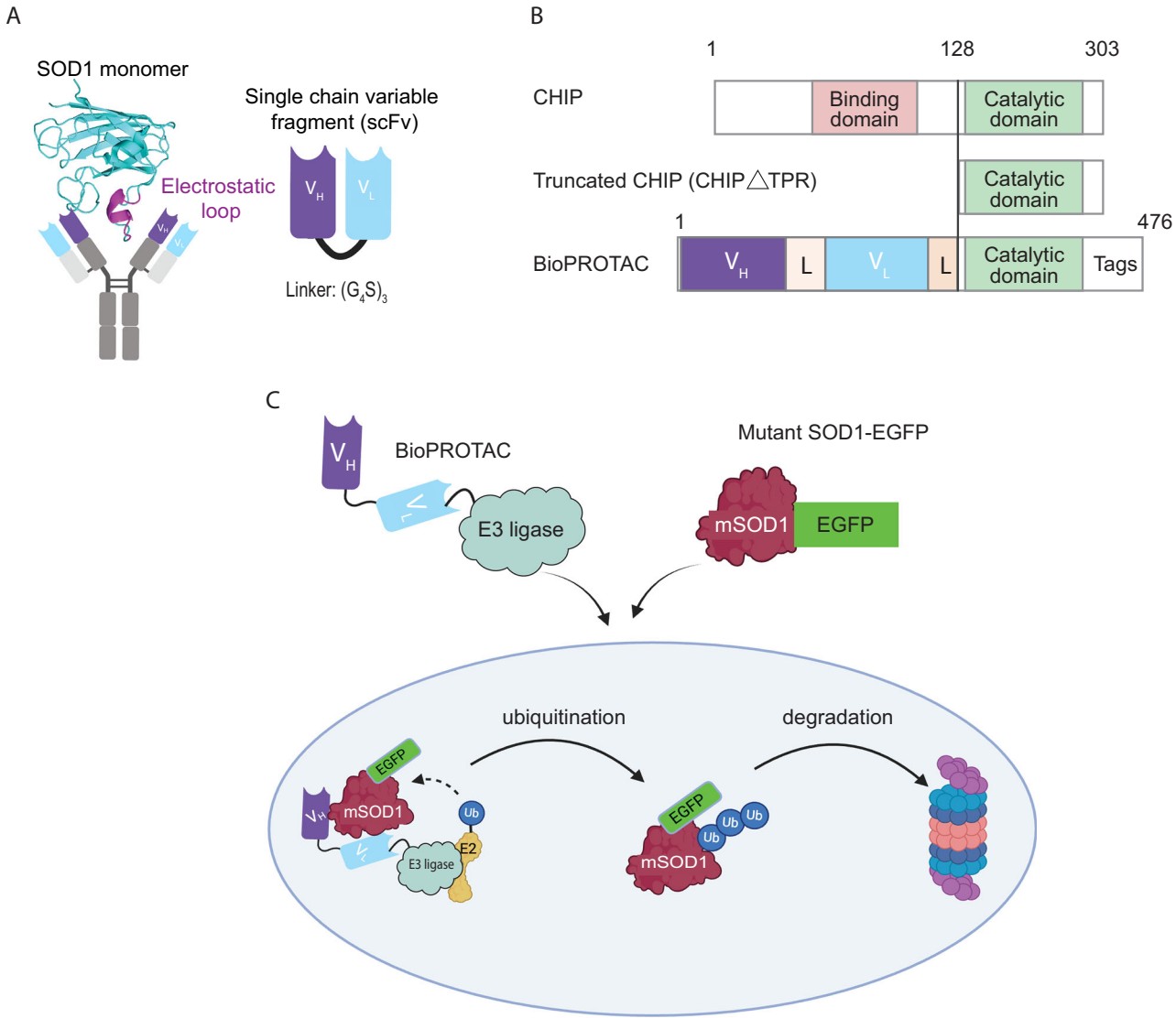

**Fig. 1 | Design and proposed mechanistic action of the SOD1-targeting Bio-PROTACs. A** Antibodies were raised against the SOD1 electrostatic loop (magenta) and single-chain variable fragments (scFvs) were generated from the antibody sequences, with heavy ($V_H$) and light ($V_L$) chains joined by a flexible $(G_4S)_3$ motif. Created in BioRender. Chisholm, C. (2025) https://BioRender.com/mg3bqw4. **B** For the initial scFv screen, the E3 ligase CHIP was truncated to remove the binding domain, and the catalytic domain was then fused to misfolded SOD1 scFvs. FLAG and 6x His tags were added for detection. **C** BioPROTACs are proposed to bind misfolded SOD1 (mSOD1) via their scFv domain, resulting in ubiquitination by the E3 ligase domain and subsequent proteasomal degradation of misfolded SOD1, which can be measured through EGFP-mediated fluorescence. Created in BioRender. Chisholm, C. (2025) https://BioRender.com/w5c64g2.

protein. Hence, BP2 was selected as the lead candidate to progress for further BioPROTAC optimisation, and used in subsequent assays and experiments described below.

## BioPROTAC-mediated reduction of mutant SOD1 is E3 ligase-dependent

The UPS-mediated mechanism of protein degradation harnessed by both PROTAC and BioPROTAC approaches requires ubiquitination of the target protein by an E3 ligase. To optimise the degradation of misfolded SOD1 observed in the initial scFv screen, we tested a panel of eight E3 ligases from across the four family groups (U-BOX, HECT, RING and Ring between Ring). Selection of candidates was based on strong expression levels in relevant cell lines and tissue types, and retention of ubiquitination activity when truncated[38–45]. The catalytic domains of selected E3 ligases, UBE4A, UBOX5, NEDD4L, UBR5, RNF4, βTrCP and Parkin (with intrinsic substrate binding domains removed), were fused to BP2, the most effective scFv from the initial BioPROTAC

screen. We configured the scFv and detection tags of each chimera to preserve the relative position of the catalytic domain relative to the rest of the protein in the native E3 ligase (Fig. 3A).

BioPROTAC candidates containing RNF4, UBE4A, Parkin, and CHIP catalytic domains showed variable soluble expression in HEK293 cells, with UBE4A and CHIP constructs displaying the highest expression (Fig. 3B). BioPROTACs containing βTrCP, UBOX5, NEDD4L and UBR5 catalytic domains were only detected in the insoluble fraction (Fig. 3C). Interestingly, all soluble BioPROTAC constructs had the scFv in an N-terminal configuration relative to the ligase domain. Immunostaining confirmed that soluble BioPROTAC ligases were highly expressed, with significantly lower expression observed with insoluble ligases (Fig. 3D). These soluble and relatively highly expressed BioPROTACs (i.e. CHIP, UBE4A, Parkin and RNF4) significantly reduced levels of misfolded SOD1$^{G93A}$-EGFP in HEK293 cells compared to control (Fig. 3E). Moreover, the BioPROTACs containing CHIP, UBE4A and Parkin significantly reduced the number of cells with insoluble

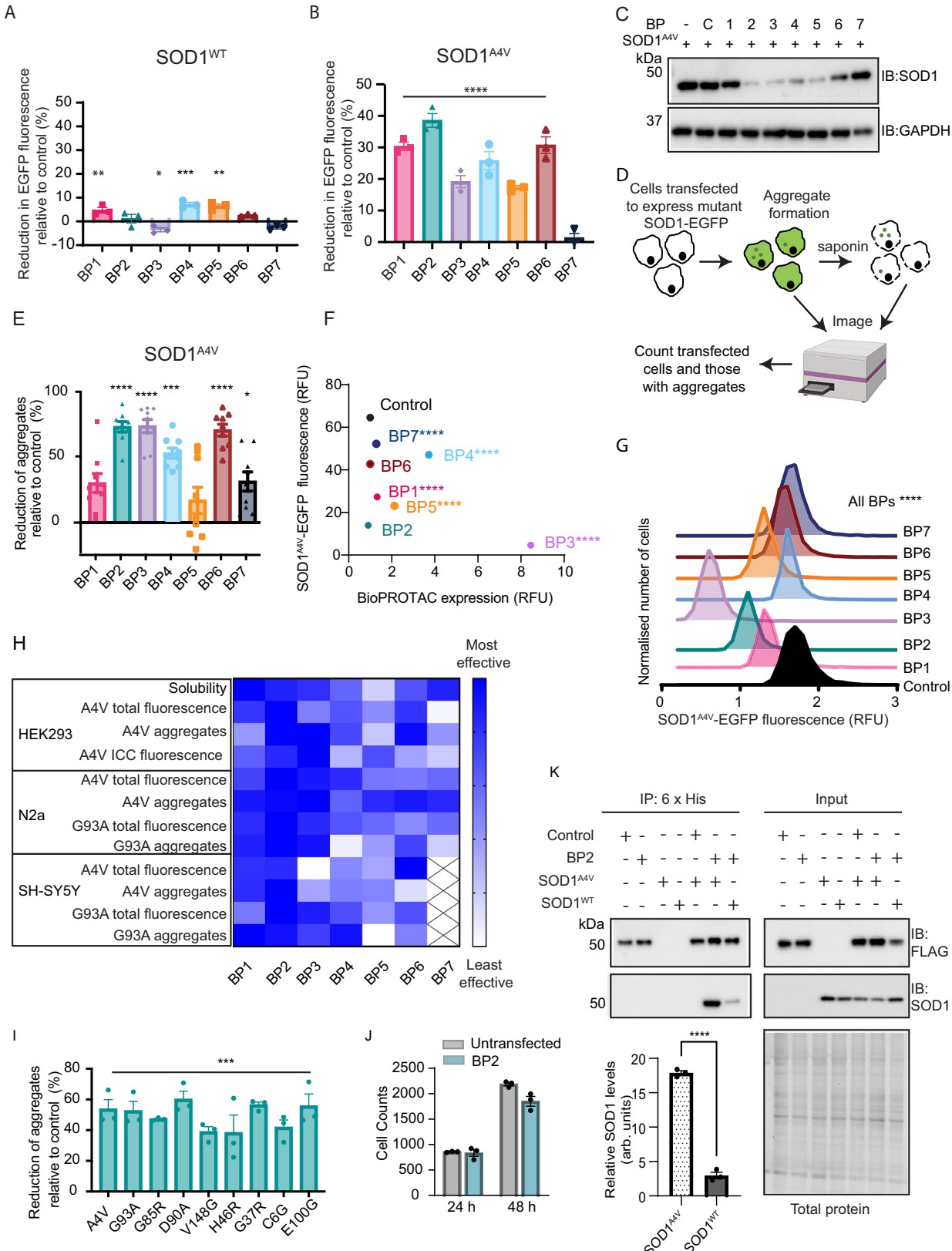

SOD1$^{G93A}$-EGFP aggregates (Fig. 3F). CHIP and Parkin also showed significant reduction of SOD1$^{G93A}$-EGFP in HEK293 cells compared to control by immunoblot (Fig. 3G). Similar effects were observed using an alternate SOD1 mutant, SOD1$^{A4V}$ (Supplementary Fig. 3A–C), and in a second cell line, SH-SY5Y (Supplementary Fig. 3D, E), with BioPROTACs containing CHIP, UBE4A and Parkin reducing levels of misfolded SOD1

compared to control. CHIP-, UBE4A- and Parkin-containing BioPRO-TACS displayed comparable capacity to reduce levels and aggregation of misfolded SOD1 across these assays (Supplementary Fig. 3F). Considering these data, and the high expression levels of CHIP in spinal cord tissue[44,45], the BioPROTAC containing BP2 and CHIP catalytic domain was selected as the lead candidate for further testing.

**Fig. 2 | BioPROTAC scFvs engage misfolded SOD1, leading to a decrease in SOD1 aggregation.** Reduction of **A** SOD1$^{WT}$-EGFP and **B** SOD1$^{A4V}$-EGFP fluorescence in HEK293 cells expressing BioPROTACs relative to cells co-transfected with the control over 48 h. **C** Immunoblot of lysates from cells transfected with SOD1$^{A4V}$-EGFP and the BioPROTAC panel. **D** Schematic representation of saponin permeability assay for measuring proportion of cells with aggregates. Created in BioRender. Chisholm, C. (2025) https://BioRender.com/vx24bql. **E** The number of cells containing insoluble SOD1$^{A4V}$-EGFP aggregates in HEK293 cells expressing BioPROTACs relative to cells co-transfected with the control was quantified using the saponin permeability assay. **F** Immunocytochemistry was used to assess BioPROTAC and SOD1$^{A4V}$-EGFP expression in HEK293 cells. **G** The reduction in SOD1$^{A4V}$-EGFP fluorescence compared to control was determined when expression levels of the BioPROTACs were normalised. **H** A weighted heat map comparing the relative efficacy of BioPROTACs in the various assays, with BP2 identified as the most

effective BioPROTAC. **I** The reduction in the number of cells containing insoluble SOD1-EGFP aggregates following expression of BP2 across a range of SOD1 variants relative to cells co-transfected with the control. **J** Cell counts for HEK293 cells untransfected or transfected with BP2 for 24 or 48 h. **K** Co-immunoprecipitation was used to assess the specificity of BP2 binding for misfolded SOD1 over the WT form. For all graphs, bars represent mean ± SEM (* $P < 0.05$, ** $P < 0.01$, *** $P < 0.001$, **** $P < 0.0001$). Statistical significance was determined using (**A**, **B**) two-way ANOVA paired with Dunnett's multiple comparisons test, (**E**, **I**) ordinary one-way ANOVA paired with Dunnett's multiple comparisons, (**F**, **G**) Kruskal–Wallis one-way ANOVA paired with Dunn's multiple comparisons test or (**J**, **K**) unpaired Student's t-tests. Blots are representative from at least 3 independent experiments. Raw data, complete western blots, total protein images and exact P-values are shown in Source data file and Supplementary Table 2.

## BioPROTACs function via both lysosomal and proteasomal degradation pathways

To investigate the E3 ligase dependency of BioPROTAC-mediated degradation of misfolded SOD1, we compared the efficacy of the BP2-CHIP chimera with a construct containing only the scFv component of BP2 (BP2$^{ΔCHIP}$). HEK293 cells were co-transfected to express SOD1$^{G93A}$-EGFP and either control, BP2 or BP2$^{ΔCHIP}$. The active BP2 chimera reduced the proportion of cells containing SOD1$^{G93A}$-EGFP aggregates compared to control (45 ± 10%), while BP2$^{ΔCHIP}$ had no effect, indicating the importance of E3 ligase activity for degradation of misfolded SOD1 (Fig. 4A). Interestingly, while the BP2 chimera reduced the number of cells containing aggregates of the more aggregation-prone mutant SOD1$^{A4V}$-EGFP (58 ± 0.1% reduction compared to control), some low-level activity was still observed in the absence of the catalytic domain, with BP2$^{ΔCHIP}$ reducing aggregation of SOD1$^{A4V}$-EGFP, by 20 ± 3% (Fig. 4A).

The effect of BP2 on misfolded SOD1 degradation was concentration-dependent. HEK293 cells were co-transfected to express SOD1$^{A4V}$-EGFP with increasing levels of either BP2 or BP2$^{ΔCHIP}$, and lysates were collected for immunoblotting. The presence of the ligase in BP2 resulted in greater reductions in cellular SOD1$^{G93A}$-EGFP levels compared to BP2$^{ΔCHIP}$ at all levels tested. Only the highest amount of BP2$^{ΔCHIP}$ reduced SOD1$^{G93A}$-EGFP levels compared to control (Fig. 4B). To investigate whether the presence of truncated CHIP in the BioPROTAC affected its endogenous function, we assessed the levels of a known CHIP substrate Hsp70 (HSPA1A) in cells expressing BP2 and BP2$^{ΔCHIP}$, and found no difference (Fig. 4B). These data suggest that endogenous substrates of CHIP are not targeted by BP2. Immunoblotting and an SOD1 activity assay confirmed the specificity of BP2 for the misfolded form of SOD1 over the WT form (Supplementary Fig. 4).

It has been proposed that scFvs that target misfolded SOD1 function by binding and preventing aggregation without reducing the overall level of misfolded SOD1, rather than degrading it[46]. However, a recent study using an scFv against α-synuclein showed that an scFv-α-synuclein complex was processed through the lysosomal degradation pathway[47]. To investigate the involvement of the two major protein degradation pathways in BP2 and BP2$^{ΔCHIP}$ action, we used a pharmacological approach. Treatment with proteasome inhibitor MG132 resulted in increased aggregate formation when cells expressed BP2 but not BP2$^{ΔCHIP}$ (Fig. 4C), suggesting the UPS plays a major role in the degradation mechanism when the E3 ligase domain is present. Treatment with Bafilomycin A (Baf.A1), a small molecule that inhibits autophagosome-lysosome fusion and acidification[48], resulted in a partial blocking of effect for cells expressing BP2, but complete loss of effect for cells expressing BP2$^{ΔCHIP}$ (Fig. 4D). Similarly, SOD1$^{A4V}$ clearance was abolished when HEK293 cells transfected with BP2 were exposed to MG132 but not when treated with Baf.A1 (Fig. 4E). Taken together, these results suggest that BP2$^{ΔCHIP}$ decreases SOD1 levels via lysosome-mediated degradation, whilst BP2 harnesses both the

lysosomal and UPS pathways to degrade SOD1, leading to improved efficacy of BP2.

## Expression of MisfoldUbL in neurons prevents weight loss and slows disease progression in SOD1$^{G93A}$ mice

Having established that a BioPROTAC composed of scFv2 and CHIP (BP2) was the most effective at degrading misfolded SOD1 with minimal off-target effects, we moved to investigate in vivo efficacy. Henceforth, we refer to BP2 as MisfoldUbL, to reflect the targeting of misfolded SOD1 via the ubiquitin ligase pathway. We generated a transgenic mouse employing the human synapsin 1 promoter to restrict expression to neurons only[49–51]. Immunohistochemistry and immunoblotting showed MisfoldUbL expression in both brain and spinal cord, but not liver (Fig. 5A, B). There was a 26-fold higher MisfoldUbL level in the brain than spinal cord (Fig. 5C).

To investigate the effect of MisfoldUbL expression on the ALS phenotype observed in SOD1$^{G93A}$ mice, we crossbred MisfoldUbL mice to generate SOD1$^{G93A}$/MisfoldUbL mice (Fig. 5D). SOD1$^{G93A}$ mice have previously shown sex-specific differences in disease phenotype[52–54], and we therefore assessed for sex-specific phenotypic changes. Male SOD1$^{G93A}$/MisfoldUbL mice displayed a significant increase in body weight compared to SOD1$^{G93A}$/WT mice (5.5 ± 0.16%) (Fig. 5E). No significant difference in body weight was observed between SOD1$^{G93A}$/MisfoldUbL and SOD1$^{G93A}$/WT female mice (Fig. 5F). Both male and female WT/MisfoldUbL mice showed a small but significant decrease in body weight compared to WT/WT mice (1.0 ± 0.07% males, 1.3 ± 0.08% females) (Supplementary Fig. 5A, B).

In the SOD1$^{G93A}$ mouse model, motor function is a primary measure for disease progression and severity, and was assessed weekly using an accelerating rotarod. SOD1$^{G93A}$/MisfoldUbL male mice exhibited a longer latency to fall than SOD1$^{G93A}$/WT mice (Fig. 5G). In contrast, female SOD1$^{G93A}$/MisfoldUbL showed a shorter latency to fall than SOD1$^{G93A}$/WT mice (Fig. 5H). For the control groups, as expected, both male and female WT/MisfoldUbL mice showed no significant difference in rotarod performance compared to WT/WT mice (Supplementary Fig. 5C, D).

While male SOD1$^{G93A}$/MisfoldUbL mice showed no significant difference in disease onset compared to SOD1$^{G93A}$/WT mice, longitudinal analysis of neurological score showed that SOD1$^{G93A}$/MisfoldUbL mice exhibited a significantly slower disease progression (Fig. 5I). Female SOD1$^{G93A}$/MisfoldUbL mice showed a delay in disease onset and a slower disease progression compared to SOD1$^{G93A}$/WT mice (Fig. 5J).

Kaplan-Meier analysis showed no difference in survival between SOD1$^{G93A}$/MisfoldUbL and SOD1$^{G93A}$/WT for either male or female mice (Fig. 5K, L). However, the similarity in survival time masked an interesting observation that became strikingly evident as the study progressed. Of the male SOD1$^{G93A}$/MisfoldUbL mice, 43% were euthanised at a defined ethical endpoint (based on weight loss) with a neurological score of 1 or 2. Even as these mice reached their weight

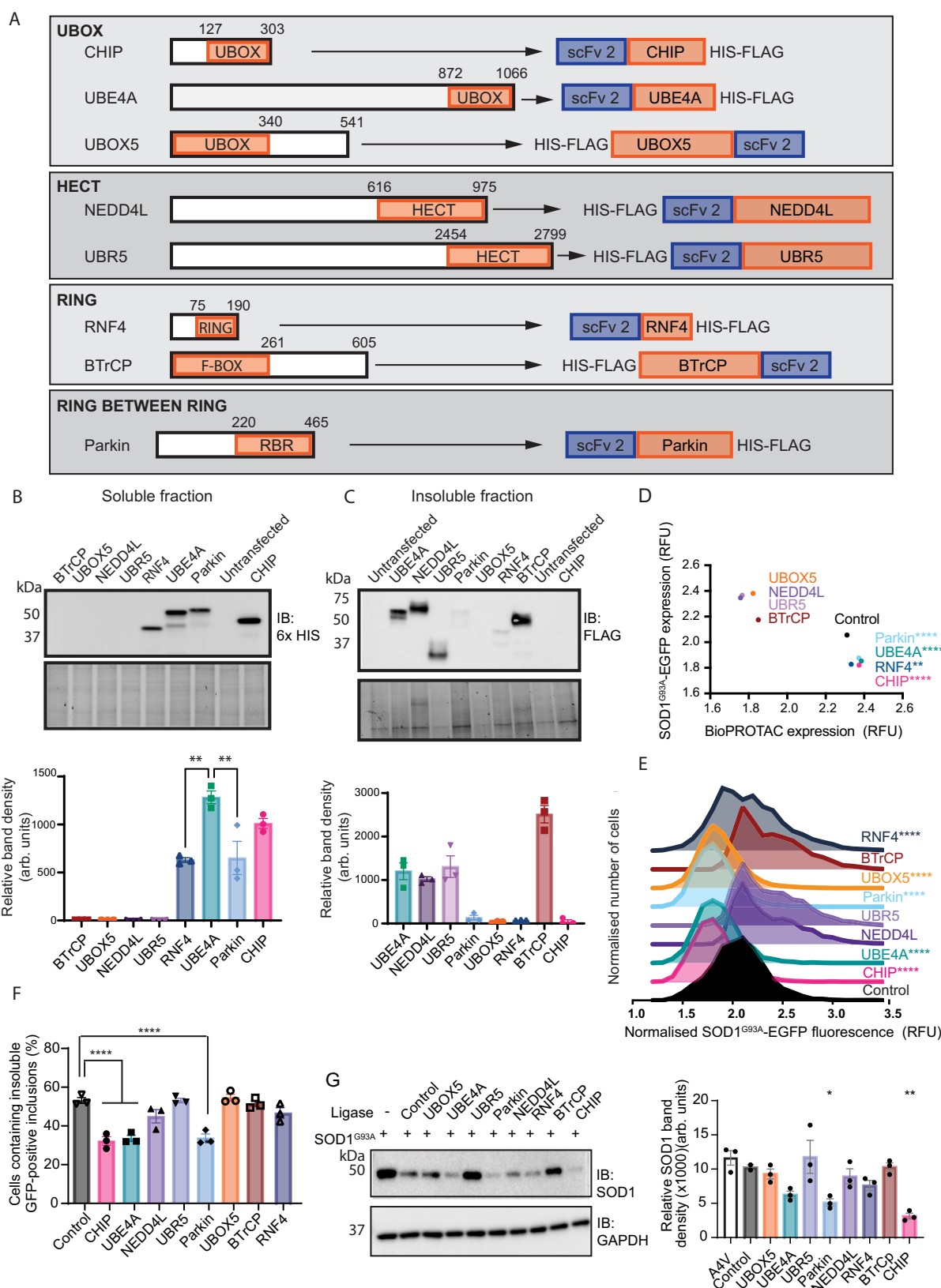

endpoint, they remained extremely mobile and continued to display behaviours that required considerable limb strength, such as rearing and hanging from the wire food hopper. SOD1$^{G93A}$/MisfoldUbL mice were also inquisitive and active in the cage (see Supplementary Videos 1 and 2 for video of SOD1$^{G93A}$/MisfoldUbL and SOD1$^{G93A}$/WT, respectively). No male SOD1$^{G93A}$/MisfoldUbL mouse reached

complete paralysis (neurological score of 4). A contingency analysis using two-sided Fisher's exact test was performed to examine the relationship between genotype and the underlying reason for endpoint (i.e. paralysis or weight loss). The difference between these variables was significant for males, with SOD1$^{G93A}$/WT mice more likely to reach paralysis (ALS 4) than SOD1$^{G93A}$/MisfoldUbL mice. A

**Fig. 3 | BioPROTAC efficacy is dependent on the E3 ligase component. A** A panel of eight E3 ligases were fused to the most effective scFv determined previously. The panel included representatives from the 4 families of ligases. Positioning of scFv and detection tags were dependent on position of the catalytic domain in the native E3 ligase. Levels of the BioPROTACs were assessed in the **B** soluble and **C** insoluble fractions from HEK293 lysates. **D** Immunocytochemistry was used to assess Bio-PROTAC and SOD1$^{G93A}$-EGFP expression in HEK293 cells. **E** The reduction in SOD1$^{G93A}$-EGFP fluorescence compared to control was determined when expression levels of the BioPROTACs were normalised. **F** The number of cells containing insoluble SOD1$^{G93A}$-EGFP aggregates in HEK293 cells expressing BioPROTACs

relative to cells co-transfected with the control was quantified using the saponin permeability assay. **G** Immunoblot of lysates from cells transfected with SOD1$^{G93A}$-EGFP and the ligase panel. For all graphs, bars represent mean ± SEM (** $P < 0.01$, **** $P < 0.0001$). Statistical significance was determined using (**B**, **C**) ordinary one-way ANOVA paired with Tukey's multiple comparisons test, (**D**, **E**) Kruskal–Wallis one-way ANOVA paired with Dunn's multiple comparisons test or (**F**, **G**) ordinary one-way ANOVA paired with Dunnett's multiple comparisons test. Blots are representative from at least 3 independent experiments. Raw data, complete western blots, total protein images and exact P-values are shown in Source data file and Supplementary Table 2.

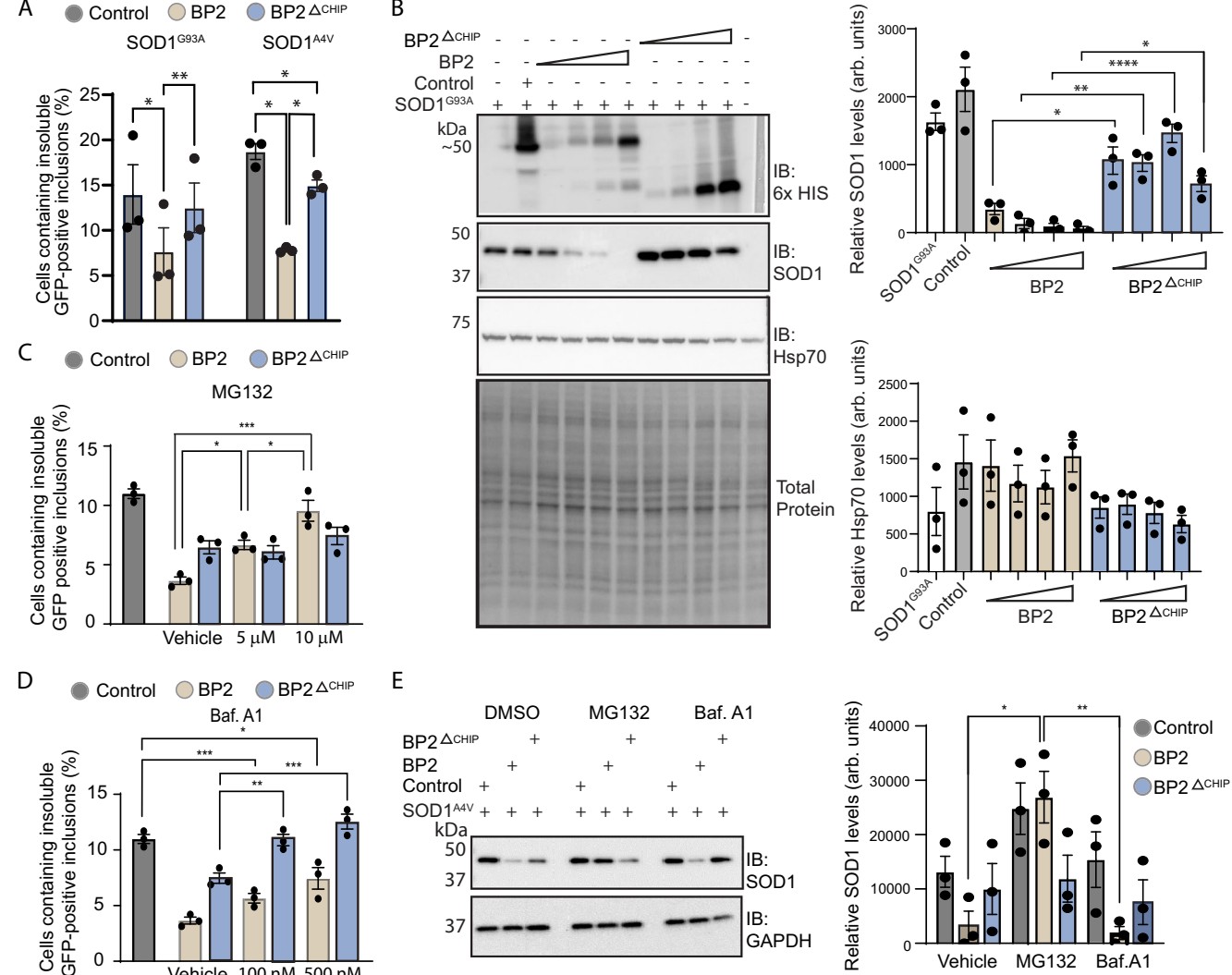

**Fig. 4 | BioPROTACs degrade misfolded SOD1 via both lysosomal and proteasomal degradation pathways. A** The number of cells containing insoluble SOD1-EGFP aggregates in HEK293 cells expressing either BP2 or BP2$^{\Delta CHIP}$ and SOD1$^{G93A}$-EGFP or SOD1$^{A4V}$-EGFP relative to cells co-transfected with the control. **B** Levels of SOD1 and Hsp70 in cells transfected with increasing amounts of BP2 or BP2$^{\Delta CHIP}$. **C** The number of cells with insoluble SOD1$^{A4V}$-EGFP aggregates after increasing concentrations of MG132 was quantified for BP2 and BP2$^{\Delta CHIP}$. **D** The number of cells with insoluble SOD1$^{A4V}$-EGFP aggregates after increasing concentrations of Baf. A1

was quantified for BP2 and BP2$^{\Delta CHIP}$. **E** Levels of SOD1 in cells transfected with BP2 or BP2$^{\Delta CHIP}$ then treated with vehicle, MG132 or Baf.A1. For all graphs, bars represent mean ± SEM (* $P < 0.05$, ** $P < 0.01$, *** $P < 0.001$, **** $P < 0.0001$). Statistical significance was determined using (**A**) repeated measures or (**B**–**D**) ordinary one-way ANOVA paired with Tukey's multiple comparisons test. Blots are representative from at least 3 independent experiments. Raw data, complete western blots images and exact P-values are shown in Source data file.

similar trend was observed with female mice, where 83% of SOD1$^{G93A}$/MisfoldUbL mice were euthanised for weight loss compared to 50% of the SOD1$^{G93A}$/WT mice. Of the female SOD1$^{G93A}$/MisfoldUbL mice that were euthanised for reaching the weight endpoint, 33% had ALS

scores of 1 or 2; however, these differences did not reach statistical significance.

In summary, these results indicate that expression of the Mis-foldUbL transgene in SOD1$^{G93A}$ mice delays disease progression and

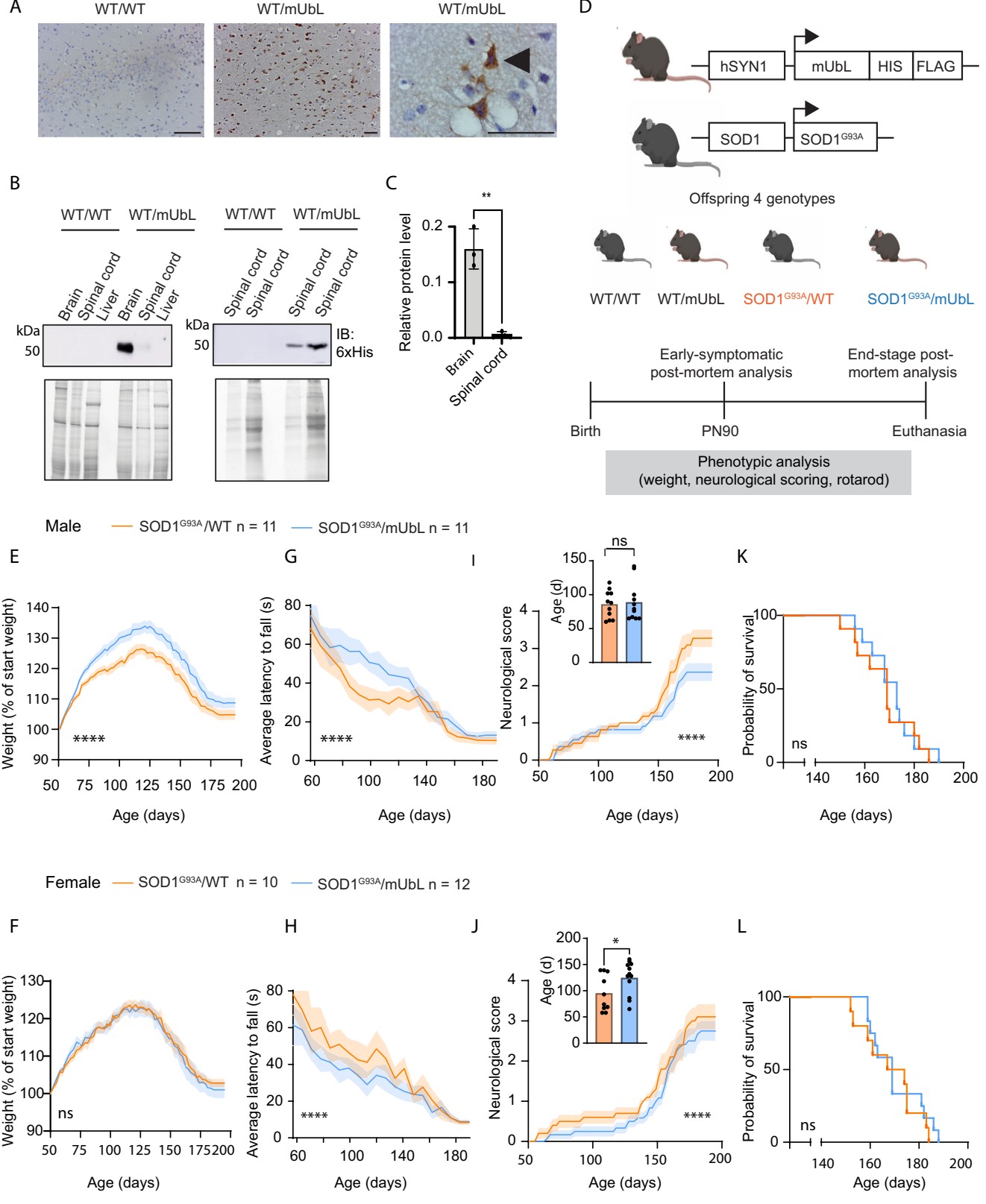

protects against progression of ALS symptoms, and this effect is more marked in male than female mice. Over time, the SOD1$^{G93A}$/MisfoldUbL mice lost comparable body weight to SOD1$^{G93A}$ mice, resulting in no difference in survival data. However, expression of MisfoldUbL preserved motor control and function until the end stage of disease.

**Expression of MisfoldUbL in brain and spinal cord neurons results in a reduction of insoluble SOD1, protection of motor neurons and a preservation of neuromuscular junctions**

To investigate underlying ALS biochemical and physiological parameters associated with the observed phenotypic effects of MisfoldUbL expression at an early-symptomatic stage, a group of age-matched

**Fig. 5 | MisfoldUbL is expressed in transgenic mice and affects phenotypic characteristics in SOD1$^{G93A}$ mice. A** Immunohistochemistry to assess expression of MisfoldUbL in neuronal cells in brain tissue of WT/mUbL mice. Images are representative of n = 6 mice. Scale bar represents 50 μm and MisfoldUbL is indicated with black arrows. **B** Immunoblot to assess expression of MisfoldUbL in brain, spinal cord and liver in WT/mUbL mice. **C** Relative levels of MisfoldUbL in brain versus spinal cord. **D** Schematic representation of experimental setup. Heterozygous SOD1$^{G93A}$ mice and heterozygous MisfoldUbL mice were crossed to obtain four experimental groups; non-transgenic WT/WT, WT/MisfoldUbL, SOD1$^{G93A}$/WT and SOD1$^{G93A}$/MisfoldUbL mice. Created in BioRender. Chisholm, C. (2025) https://BioRender.com/zjatewp. **E–L** Phenotypic data comparing SOD1$^{G93A}$/WT (n = 11 male

and n = 10 female) and SOD1$^{G93A}$/MisfoldUbL mice (n = 11 male and n = 12 female) are shown. Data of control groups WT/WT (n = 24) and WT/MisfoldUbL (n = 24) are shown in Supplementary Fig. 5. Male and female mice were assessed for **E, F** weight gain, **G, H** motor function via latency to fall on the rotarod, **I, J** disease progression via neurological score and **K, L** survival. Results represent mean ± SEM (shading) (* $P < 0.05$, ** $P < 0.01$, **** $P < 0.0001$). Statistical significance was determined using (**C, I** inset and **J** inset) unpaired, two-tailed t-tests, (**E–J**) repeated measures one-way ANOVA paired with Tukey's multiple comparisons test or (**K, L**) log-rank Mantel-Cox test. Blots are representative from at least 3 independent experiments. Raw data, complete western blots, total protein images and exact P-values are shown in Source data file.

mice were sacrificed at 90 days old. Immunoblot analysis of soluble and insoluble fractions from brain and lumbar spinal cord homogenates showed a decrease in insoluble SOD1 in brain tissue at both early-symptomatic stage (Fig. 6A) and end stage (Fig. 6B) in SOD1$^{G93A}$/MisfoldUbL mice compared to SOD1$^{G93A}$/WT mice. However, this reduction in insoluble SOD1 was not observed in the spinal cord in either cohort, despite expression of the MisfoldUbL across all regions of the spinal cord (Supplementary Fig. 5E). There was no difference in soluble SOD1 in either cohort in the brain or spinal cord.

To investigate the retention of muscle function observed at end stage in the SOD1$^{G93A}$/MisfoldUbL mice, we performed a motor neuron count in the lumbar spinal cord (Fig. 6C) and measured neuromuscular junction innervation in gastrocnemius muscles (Fig. 6D), and found the MisfoldUbL transgene conferred protection of both integral components of motor function. SOD1$^{G93A}$/MisfoldUbL mice exhibited a 30 ± 6% increase in motor neuron number compared to SOD1$^{G93A}$/WT mice at early-symptomatic stage, and 50 ± 15% at end stage. For the WT/MisfoldUbL and WT/WT control groups, there was no significant difference in motor neuron number at either the early-symptomatic stage or when the mice had reached 200 days old (Supplementary Fig. 5F). SOD1$^{G93A}$/MisfoldUbL mice also displayed an increase in the number of fully innervated neuromuscular junctions in their gastrocnemius muscles at end stage. These findings are consistent with the functional observations noted for these mice, and further support the neuroprotective effect conferred by expression of MisfoldUbL.

## Discussion

The aberrant aggregation of proteins is implicated in the onset and pathogenesis of a wide range of neurodegenerative disorders, including ALS. Increasing evidence indicates that misfolded protein oligomers produced as intermediates in the aggregation process are potent neurotoxic agents in disease[55], however, the heterogeneous and transient nature of these misfolded species have made it challenging to develop effective therapeutics. Gene silencing via antisense oligonucleotides[56–59] and RNA interference[60,61] are current state-of-the-art therapeutic technologies; however, the lack of specificity to the toxic form of the protein in these silencing approaches is a major limitation in diseases where the correctly folded and functioning form of the protein is integral to cellular health. Targeted protein degradation, specifically of the pathogenic form of the protein, may provide an innovative alternative strategy, maintaining native forms of the protein.

Here we report a genetic-based targeted protein degradation strategy that effectively reduces both soluble and aggregated misfolded SOD1 whilst preserving the natively folded WT form. Misfolded SOD1 antibodies[14,30] provide the specificity critical to our BioPROTAC design, and were effective across a range of disease-associated conformers. Interestingly, several E3 ligases screened in this study were not effective, despite displaying efficacy in previous BioPROTAC designs[38,62,63]. This highlights the importance of evaluating PROTAC components in disease-relevant assays. With over 700 potential E3 ligases[64,65], many still unidentified, there is huge untapped potential in

this area. In our BioPROTAC design, the chaperone-associated E3 ligase CHIP proved most effective, potentially reflecting its existing endogenous role as a mediator of degradation for neurodegenerative disease-associated proteins[66–71].

In the SOD1$^{G93A}$ mouse model, the presence of the BioPROTAC transgene, MisfoldUbL, resulted in a delay in disease onset, a slowing of disease progression and a retention of motor function at endpoint. These phenotypic observations correlated with a reduction of insoluble SOD1 in the brain, and the preservation of motor neurons in the ventral lumbar spinal cord and innervated neuromuscular junctions. Despite these promising results, the protective effect of MisfoldUbL did not extend survival in the ALS mouse model of disease. This may be due to the choice of promoter used to drive expression of the transgene in this work. The human synapsin 1 (h*SYN1*) promoter has been used extensively for neuron-specific expression of proteins in mouse models[49,50,72–74]. However, in comparison with ubiquitous promoters, the synapsin promoter drives lower levels of expression[50] and, in our constitutive expression model, levels of MisfoldUbL were notably lower in the spinal cord, a critical site of pathology for ALS, than the brain. As the SOD1$^{G93A}$ mouse model overexpresses mutant SOD1 by a factor of >20[75], broader and stronger expression of MisfoldUbL may be required to combat the persistence of disease phenotype and highlights an important next step to investigate this strategy in patient-derived cell lines that may capture endogenous expression levels. There is also increasing evidence that sex plays a significant role in ALS in both humans[76] and the SOD1$^{G93A}$ mouse model[77,78]. In this study, whilst we found that MisfoldUbL conferred protection to both sexes, the impact was greatest for male mice. The physiological reasons underlying sex-specific differences in ALS are yet to be elucidated.

Small-molecule PROTACs have been the subject of intense research over the past two decades, with 34 reaching clinical trials in the past 5 years[79], although only one of which is for a neurodegenerative disease-associated protein. This highlights the additional challenges faced by targeting proteins in the CNS. Small-molecule PROTACs have several disadvantages in the context of many neurodegenerative disease-associated proteins; they rely on a structurally defined, comparatively rigid binding pocket on the target protein, they are susceptible to off-target effects, and they require careful investigation to ensure substrate specificity between ligands and E3 ligases/protein of interest[80]. Furthermore, they demonstrate a hook effect where saturating doses of PROTAC reduce degradation efficiency[81]. In contrast, biological PROTACs comprising of a genetic fusion of protein-binder to truncated E3 ligase specifically bind the target protein with higher affinity than small molecules, do not require a binding pocket, can recognise a misfolded form of a protein over the WT form, are less demanding to design and synthesise, and display superior safety and less toxicity (reviewed in refs. 82,83). Nevertheless, rigorous validation of specificity and functional consequences is essential in the development of safe and effective biological PROTAC therapies.

One limitation of using antibodies as the binding domain in a BioPROTAC design applies to mutations that lie within the epitope, including truncation mutations that remove the epitope completely. It

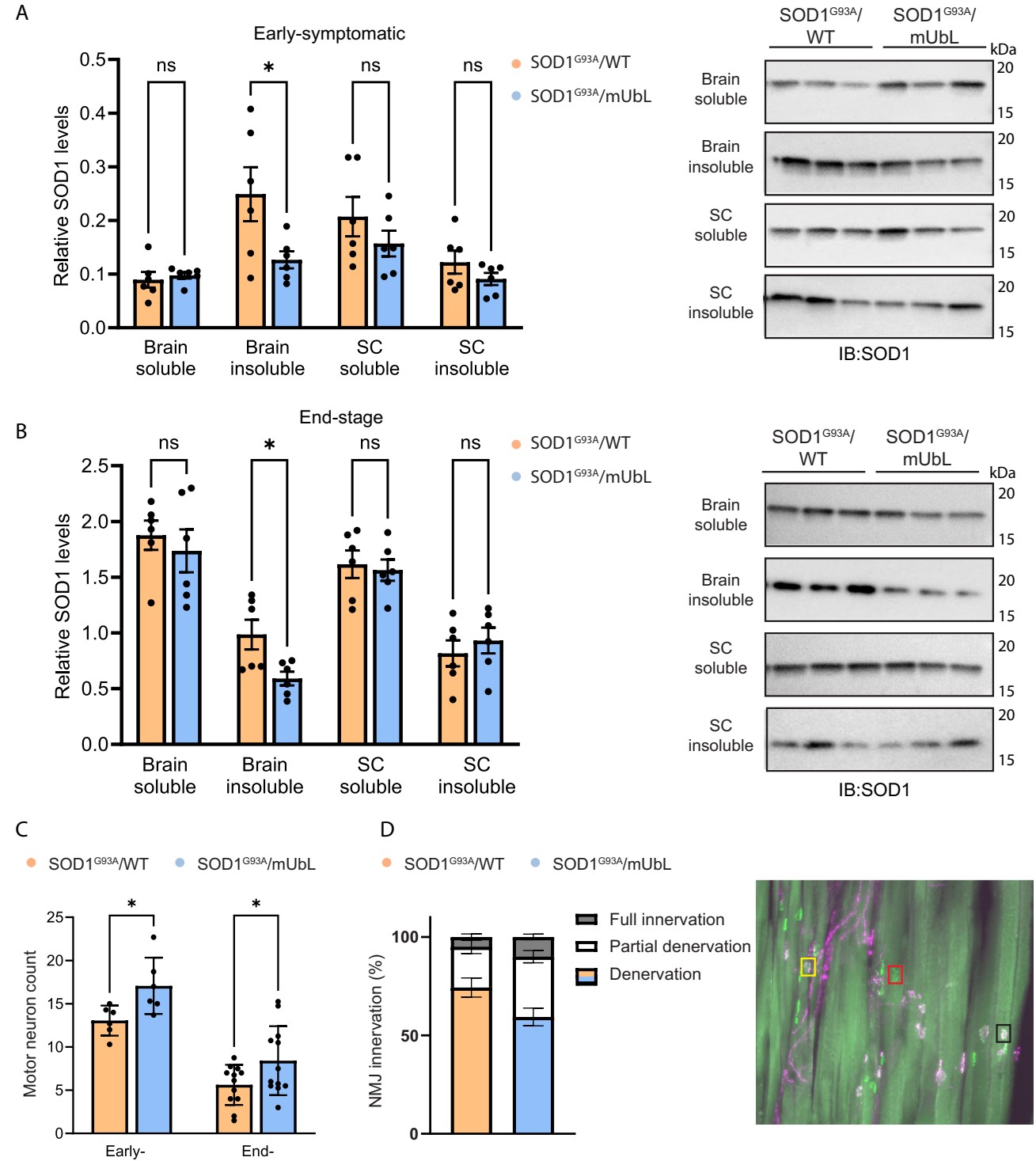

**Fig. 6 | Expression of MisfoldUbL in brain and spinal cord neurons results in a reduction of insoluble SOD1, protection of motor neurons and a preservation of neuromuscular junctions in SOD1$^{G93A}$ mice.** Immunoblots to assess levels of soluble and insoluble SOD1 protein in the brains and spinal cords of **A** early-symptomatic and **B** end-stage SOD1$^{G93A}$/MisfoldUbL and SOD1$^{G93A}$/WT mice. **C** Motor neuron numbers were counted in the ventral lumbar spinal cord of early-symptomatic and end-stage mice. **D** The number of fully innervated, partially innervated and denervated neuromuscular junctions of SOD1$^{G93A}$/WT and SOD1$^{G93A}$/MisfoldUbL mice. A representative image of a stained longitudinal gastrocnemius section is included showing the pre- (purple) and post-synaptic (green) markers. Scale bar represents 100 μm. A fully innervated NMJ is indicated in a solid black box, a partially innervated NMJ in a dashed yellow box, and a denervated NMJ in a dotted red box. Results represent mean ± SEM, n = 6 (early-symptomatic cohort) or 12 (end-stage cohort) mice per genotype (* $P < 0.05$). **A–D** Statistical significance was determined using unpaired two-tailed t-tests. Raw data, complete western blots images and exact P-values are shown in Source data file.

is possible that tailoring the antibody to the mutation of the patient in a personalised therapy, as has been proposed for Alzheimer's treatment[84], may alleviate this limitation. Furthermore, advances in artificial intelligence may also provide the design of personalised antibodies that have greater binding affinity to a target than exist in nature[85,86].

In conclusion, we show that a BioPROTAC approach is a promising therapeutic strategy to reduce pathological protein in a neurodegenerative disease context. The transition of this proof-of-concept study into a viable treatment strategy for SOD1-ALS patients and its application to other neurodegenerative disease-associated misfolded proteins warrants further research.

## Methods

### Plasmid construction
Plasmids encoding different mutations of SOD1 with a C-terminal EGFP tag in a pEGFP-N1 backbone have been previously described[87,88]. BioPROTAC plasmids were designed using the plasmid backbone from the expression vector pcDNA3-R4-uAb, containing an scFv for β-galactosidase fused to a truncated version of the C-terminus of Hsc70-interacting protein (CHIPΔTPR). This plasmid served as the control BioPROTAC for all experiments. For the scFv panel, the scFv for β-galactosidase was removed and replaced with sequences for misfolded SOD1 scFvs. For the ligase panel, CHIPΔTPR was removed and replaced with truncated sequences for NEDD4L, parkin, UBR5, UBOX5, RNF4, BtrCP or UBE4A. For scFv plasmids alone, the CHIPΔTPR sequence was removed. The pcDNA3-R4-uAb vector was a gift from Matthew DeLisa (Addgene plasmid #1001800). All plasmids were synthesised and sequenced by Gene Universal (Newark, DE).

### Cell culture and transfection
Mouse neuroblastoma (Neuro-2a, ATCC CCL-131)[89], human neuroblastoma (SH-SY5Y, ATCC CRL-2266)[90] and human embryonic kidney (HEK293, ATCC CRL-1573)[91] cells were maintained at 37 °C in a humidified incubator with 5% atmospheric $CO_2$. Cells were transfected using the TransIT-X2 Dynamic Delivery System (Mirus Bio, Madison, WI) or Lipofectamine (Thermo Fisher Scientific, Waltham, MA), according to the manufacturer's instructions. The TransIT-X2:DNA ratios were optimised for each cell type and determined to be 2–6 μL TransIT-X2 per μg DNA. Cells were transfected when plates were ~50% confluent with 0.1, 0.5 and 2.5 μg DNA per well for 96-, 24- and 6-well plates, respectively, and 0.2 μg DNA per well for 8-well chamber slides. For co-transfections, the amount of DNA was divided equally between constructs. For co-transfections where the amount of BioPROTAC was varied, a mock plasmid was used to ensure equivalent amounts of DNA. All cell lines were verified by STR profiling and checked periodically for mycoplasma.

### Immunocytochemistry
Cells were grown in 96-well optical bottom plates or on coverslips placed in 24-well plates, and co-transfected with plasmids encoding various mutants of EGFP-tagged SOD1 and different BioPROTACs. After fixing with 4% (w/v) paraformaldehyde (PFA), cells were permeabilised with 0.1% (v/v) Triton X-100, blocked with PBS containing 10% (v/v) calf serum, 2% (w/v) bovine serum albumin (BSA) and 0.1% (v/v) Triton X-100, and probed with primary and secondary antibodies. All antibodies used in this study are listed in Supplementary Table 1.

Cells on coverslips were imaged using a Leica TCS SP8 confocal microscope with a 63× oil immersion objective lens (Leica Microsystems, Wetzler, Germany). Cell nuclei were counterstained with Hoechst nuclear stain with excitation at 405 nm and emission analysed from 420 nm to 430 nm. The EGFP fluorescent-tagged SOD1 was excited with 10% laser transmission at 488 nm, and emission was analysed from 510 to 550 nm. BioPROTACs were imaged using excitation of the Alexa Fluor 647-labelled secondary antibody at 650 nm,

with emission collected from 670 nm to 690 nm. Fluorescent emissions were acquired by sequential scanning using the Leica Application Suite - Advanced Fluorescence (LAS-AF) software (Version 3, Leica Microsystems). A minimum of 100 transfected cells were imaged across a minimum of 10 fields of view per replicate.

For analysis of SOD1-EGFP aggregates, cells in 96-well plates were imaged using a Thunder automated microscope (Leica Microsystems). Each well was imaged in a 9×9 tile scan. Images were not overlapped to prevent any duplication of counted cells impacting downstream analyses. Cells expressing SOD1-EGFP and BioPROTACs were analysed using the automated image analysis software CellProfiler (Version 4.2.4 for Windows, Broad Institute, Cambridge, Massachusetts), as described previously[92]. All images generated via automated microscopy underwent pre-processing quality control to omit out-of-focus images. After quality control processing, images were processed in CellProfiler to segment cells within the range of 30 – 130 pixel units, and measure intensity, granularity, size, shape, intensity distribution and texture. To normalise the data, the fluorescence intensity of SOD1-EGFP-positive cells was divided by the BioPROTAC fluorescence value for each cell expressing both constructs. The fluorescence intensity across all cells transfected with the control BioPROTAC was averaged, and all BioPROTAC fluorescence intensity measurements were then divided by this value to create a normalised BioPROTAC expression level. The normalised SOD1-EGFP values were then divided by the normalised BioPROTAC expression level.

### Live-cell imaging
The fluorescence of cells expressing SOD1-EGFP was monitored over a 48 h time course in an IncuCyte® ZOOM automated fluorescent microscope (Sartorius, Göttingen, Germany), as described previously[93]. Images of cells in 96-well plates were acquired every 3 h using a 10× objective in both phase and green channels, with the green channel acquisition time set at 300 ms. Cell images were analysed using a processing definition trained to select GFP-positive cells (Top-Hat segmentation, 2 Green Calibrated Unit (GCU) threshold adjustment, edge splitting, hole fill clean-up 60 μm² and minimum area filter of 200 μm²). To compare the reduction of misfolded SOD1 by the BioPROTACs, the number and fluorescence intensity of GFP-positive cells were measured in each image, normalised to the measurements at time zero, and then normalised to the cells transfected with the control BioPROTAC.

### Measurement of insoluble SOD1 aggregates and soluble misfolded SOD1
Insoluble SOD1-EGFP aggregates were calculated using a saponin-based permeabilisation protocol[94]. Saponin is a mild cholesterol-chelating detergent that creates pores in the plasma membrane of cells[95]. These pores allow soluble proteins to diffuse out of the cell, while trapping insoluble protein aggregates that are too large to fit through the pores. Cells in 96-well plates were treated with 0.03% (v/v) saponin (Sigma, Darmstadt, Germany) in PBS 48 h post-transfection. After a 10 min incubation at room temperature, plates were re-imaged and the EGFP signal analysed using a processing definition trained to select EGFP-positive cells. Immediately following saponin treatment, 100 μL of media was transferred from the top of the wells to a new 96-well plate, and EGFP soluble fluorescence measured on a POLARstar Omega plate reader (BMG Labtech, Ortenberg, Germany). Settings included a 2 × 2 matrix well scan reading from the bottom optic, with excitation at 485 nm and emission at 520 nm. For all fluorescence experiments, the mean fluorescence ± SEM was calculated across triplicate replicates.

### Degradation inhibitor assay
To investigate the mechanism of degradation, the proteasome inhibitor MG132 and the lysosome inhibitor Bafilomycin A (Baf.A1) were

used. HEK293 cells in 96-well plates were co-transfected with plasmids containing SOD1-EGFP and BioPROTACs or scFv alone, and treated for 8 h with DMSO alone, 5 or 10 μM MG132 in DMSO or 0.5 or 1 μM Baf.A1 in DMSO. Cells were then analysed for number of aggregates using the saponin assay described above.

## Immunoblotting
Transfected cells were lysed in ice-cold RIPA buffer containing Halt™ Protease Inhibitor Cocktail (Thermo Fisher Scientific), and the soluble fraction collected. The pellet was then resuspended in 10 mM Tris buffer, pH 8, containing 8 M urea and 4% (w/v) CHAPs, and the insoluble fraction collected. Lysates (30 μg total protein) were run on 4–20% Criterion™ TGX Stain-Free™ gels (Bio-Rad, South Granville, Australia), transferred to PVDF membranes (GE Healthcare, Chicago, IL), and probed with primary and secondary antibodies. For immunoblots probed with anti-6x HIS antibody, cell lysates from HEK293 cells transfected with MisfoldUbL plasmid were used as size markers instead of a protein ladder. Membranes were visualised with chemiluminescent substrate (Thermo Fisher Scientific), and imaged (Amersham Imager 600RGB, GE Healthcare), with analysis and quantification using ImageJ (Version 1.53c for Windows, National Institutes of Health, Bethesda, MD).

## Co-immunoprecipitation assay
The Dynabead™ co-immunoprecipitation kit (Thermo Fisher Scientific) was used for the detection of SOD1 binding by BioPROTACs, according to the manufacturer's instructions. Briefly, HEK293 cells in 6-well plates were transfected with BioPROTAC or SOD1-EGFP alone, or co-transfected with BioPROTAC and SOD1$^{WT}$-EGFP or SOD1$^{A4V}$-EGFP. Cells were lysed 48 h post-transfection, and lysates were mixed with 1.5 mg of magnetic dynabeads coupled to anti-6x His tag antibodies for 30 min at 4 °C. After washing, bound proteins were eluted and immunoblotted with anti-FLAG and anti-SOD1 antibodies as described above.

## In-gel zymography for SOD1 enzymatic activity
Cell lysates from transfected HEK293 cells were generated as aforementioned, with the exception that lysis buffer was 100 mM Tris-base (pH 7.5) with 0.1% TX-100 (v/v) and protease inhibitor. Cell lysates were mixed 1:2 with 3× native-PAGE sample buffer (240 mM Tris–HCl [pH 6.8], 30% glycerol [v/v], 0.03% bromophenol blue [w/v]) and loaded into Tris–glycine native-PAGE gels (4.5% stacking gel [pH 8.8], 7.5% resolving gel [pH 8.8]). Samples were electrophoresed for 30 min at 60 V and then for 2.5 h at a constant voltage of 125 V at 4 °C. Following native-PAGE, EGFP signal in the gel was detected using a ChemiDoc MP Imaging System (Bio-Rad). Gels were then subjected to zymography as described previously[96] and imaged using a GS-900 Calibrated Densitometer (Bio-Rad). Quantification of fluorescence signal and enzymatic activity was performed using ImageJ.

## Animals
The generation of transgenic mice harbouring the *misfoldUbL* transgene was performed by Genome Editing Macquarie (GEM) (Sydney, Australia), as previously described[97] with slight modification in the procedure. Briefly, cDNA for the *misfoldUbL* transgene was incorporated into an AAV plasmid downstream of the *synapsin 1* promoter, and upstream of the woodchuck hepatitis virus post-transcriptional regulatory element (WPRE), and the bovine growth hormone polyadenylation sequence (bGH pA). Homology arms (800 bp) targeting the murine *Rosa26* safe harbour locus were incorporated in this donor plasmid and loaded into AAV6 (Vector Builder Inc, Chicago, IL). AAV6 particles were electroporated (Nepa21, Nepa Gene Co, Chiba, Japan) in C57BL/6J zygotes, together with Cas9 RNPs targeting the *Rosa26* locus (sgRNA1: 5′-ACTCCAGTCTTTCTAGAAGA-3′; sgRNA2: 5′-CGCCCATCTT CTAGAAAGAC-3′; sgRNA3: 5′-CAGTCTTTCTAGAAGATGGG-3′).

Transgene-specific PCR identified 5 potential founders, and the colony was established by breeding a selected founder (F0). Inheritance was confirmed by breeding the F0 with WT C57BL/6J to produce F1 offspring. To generate the double transgenic mouse line, female MisfoldUbL mice (B6.Cg-Tg(*hsyn-mUbL*)Yer/j) were bred with male mice hemizygous for the human *SOD1$^{G93A}$* transgene maintained on a C57BL/6J background (B6-Tg (*SOD1-G93A*)1Gur/j). Breeding occurred at the Australian Bioresources Animal Facility (Moss Vale, Australia). Pups were weaned and genotyped at approximately postnatal days 21–28. Mice were genotyped by PCR using genomic DNA extracted from tail tissue. Primers were as follows:

MisfoldUbL: Fwd: ATTACGTCGACGGAGCAGAC Rev: AAGGAA GGTCCGCTGGATTG

SOD1 (human): Fwd: CATCAGCCCTAATCCATCTGA Rev: CGCGAC TAACAATCAAAGTGA

Non-transgenic WT/WT, transgenic WT/MisfoldUbL, transgenic SOD1$^{G93A}$/WT and double transgenic SOD1$^{G93A}$/MisfoldUbL mice matched for date of birth and sex were transported to the University of Wollongong (Wollongong, Australia). Mice were housed with littermates in IVC cages (Greenline GM500, Techniplast, Lane Cove West, Australia) under a 12:12 h light–dark cycle (illuminated from 0700 to 1900 h) and humidity range of 40–60%. IVC cages included a layering of iso-PADS™ bedding (Envigo, Indianapolis, IN), tissue, Bed r'Nest™ (Techniplast), a plastic house and a PVC tunnel. Food and water were available *ad libitum*. There were no significant differences between sex-matched groups at the beginning of the study (50 days of age). Once mice reached 100 days old, water-soaked food pellets were placed on the cage floor and longer sippers placed on water bottles.

## Efficacy of MisfoldUbL expression
Commencing at 50 days of age, mice (n = 12/genotype/sex, 96 total) were weighed and scored 3 days a week to assess neurological deficit, using the criteria outlined by the ALS Therapy Development Institute[98]. Scoring was performed by observers blinded to treatment.

## Rotarod
To assess the influence of MisfoldUbL on motor function, mice were assessed on a 5-lane accelerating rotarod (RotaRod Advanced, TSE Systems, Hesse, Germany). One week after arrival at the University of Wollongong, mice were habituated to the rotarod. Habituation sessions consisted of five acclimatisation sessions, with the first and second sessions run at an inclining speed of 0-4 rotations per min (rpm) for 180 s. The final three habituation sessions were performed at an inclining speed of 4–20 rpm over a 180 s period. During the recording period (testing), the rotation speed of the rotarod was 4–20 rpm over a 180 s period, with the time taken to fall (latency to fall) recorded for each mouse. Mice were given three independent runs with a 30–60 s rest between runs. The maximum time each mouse was able to remain on the rod was recorded and included in the data analysis.

## Survival and end-stage analysis
Disease end stage was defined when a mouse displayed either a 20% loss in maximum body weight and/or reached a clinical score of 4 (inability to right itself within 10 s after being placed on both sides). At this point, mice were euthanised via asphyxiation using a slow-fill carbon dioxide technique. Mice were then transcardially perfused with either PBS or 4% (w/v) PFA in PBS, and brains, lumbar spinal cords, liver and gastrocnemius muscles removed. Tissue collected from mice perfused with PBS was snap frozen in liquid nitrogen and stored at −80 °C. Tissue collected from mice perfused with 4% (w/v) PFA in PBS was post-fixed in ice-cold 4% (w/v) PFA in PBS for 24 h, and then rinsed in PBS. For spinal cords and brains, fixed tissue was then processed using a Leica ASP3000 and embedded in paraffin on a Leica EG1150 embedding centre. For gastrocnemius, fixed tissue was cryoprotected in 20% (w/v) sucrose in 0.1 M PBS for 48 h, embedded within optimal

cutting temperature (OCT) compound over liquid nitrogen, and stored at −80 °C.

## Age-matched early-symptomatic cohort

To investigate the neuropathology of mice across genotypes, an age-matched cohort (n = 10-12/ genotype) were transported to the University of Wollongong at ~80 days old, habituated for at least 7 days, and sacrificed at 90 days old as above. For this cohort, mice from each of the four genotypes were similarly split into two post-analysis groups, transcardially perfusing with either PBS or 4% (w/v) PFA for immunoblot and immunohistochemical analysis, respectively.

## Brain and lumbar spinal cord homogenisation

PBS-perfused brains and spinal cords were homogenised in PBS/ 0.01% (v/v) Triton X-100 containing 100x Halt™ protease inhibitor cocktail using a micro pestle and 5× volume of homogenisation buffer per weight of tissue (μL/mg). Homogenates were sonicated for $3 \times 5\,s$ bursts at 50% amplitude and centrifuged at $20,000 \times g$ at 4 °C for 30 min to collect soluble proteins. Insoluble proteins were extracted using an 8 M urea buffer at 2× the weight of original tissue. Detection of MisfoldUbL and SOD1 protein was performed via standard western blotting with 20 μg of loaded protein. Anti-6x His tag antibody was used to detect MisfoldUbL, and polyclonal anti-SOD1 antibody were used for SOD1 protein detection.

## Immunohistochemistry

For motor neuron counts, lumbar spinal cord transverse slices (5 μm) were stained with haematoxylin as per standard procedures and imaged using a Leica DM750 at 10× magnification. Visualisation and counts of the number of motor neurons (MN) in each section were performed using ImageJ software. Alpha motor neurons were defined as cells with an equivalent diameter of at least 20 μm[99]. Counts from the ventral horns of the right and left hemispheres were summed for each section, and then the average count from 3 sections used as a single measurement per animal for statistical analyses. MN counts were performed by two observers blind to treatment, and the counts averaged.

## Neuromuscular junction immunostaining

For neuromuscular junction staining, longitudinal gastrocnemius sections (50 μm) were bleached of background autofluorescence in PBS with a high intensity LED light at 4 °C for 48 h. Sections were then permeabilised with 2% (v/v) Triton X-100 in PBS, blocked with PBS containing 5% (v/v) normal goat serum, 1% (v/v) Triton X-100, and 3% (w/v) BSA, and then probed with rabbit polyclonal antibodies against synaptophysin and anti-neurofilament heavy polypeptide. Sections were then probed with Alexa Fluor 647-conjugated secondary antibody and Alexa Fluor 488-conjugated alpha-bungarotoxin, and mounted with ProLong Diamond Anti-fade medium (Thermo Fisher Scientific).

Stained sections were imaged using a DMi8 microscope (Leica Microsystems) at 10× magnification, with a z-stack acquisition to capture full tissue depth. Tiled images were merged, and a maximal projection image generated for analysis to maintain resolution. For each sample, at least 100 neuromuscular junctions (NMJs) were counted, with the innervation of muscles being defined by the co-localisation of fluorescently labelled post-synaptic and pre-synaptic receptors. 'Denervated', 'partially innervated', and 'fully innervated' NMJs were manually counted in ImageJ using the Cell Counter plugin, based on the presence and extent of co-localisation between the pre- and post-synaptic markers.

## Statistics

All statistical analyses were performed using GraphPad Prism software (Version 10.0 for Windows, Boston, MA). Data is presented as the mean ± SEM, unless otherwise indicated. One-way ANOVA paired with Tukey's HSD or Dunnett's multiple comparison post-test or two-way ANOVA paired with Dunnett's multiple comparison post-test were used to analyse and compare differences between multiple treatments if the data was normally distributed. Kruskal–Wallis test with Dunn's multiple comparisons post-test was used for data that did not pass the normality test. Unpaired Student's t-tests were performed for single treatment comparisons. For body weight measurements, rotarod activity and ALS score, differences between genotype groups at each recorded time-point and over the entire period were compared using repeated measures one-way ANOVA, followed by post hoc analysis using Tukey's HSD. Age of disease onset was measured as the time to reach an ALS score of 1 and, for the whole cohort, data were analysed using two-tailed Student's t-tests. To assess gender and genotype effect for age of disease onset, two-way ANOVA with Fisher's least significant difference test was performed. Survival of SOD1$^{G93A}$/MisfoldUbL and SOD1$^{G93A}$ mice were compared using log-rank (Mantel-Cox) test, with median values reported. To prevent decomposition of means due to animals reaching end-stage, for weight, score and rotarod analyses, the last values prior to euthanasia were carried forward until the last mouse in each group had reached end-stage. These values were used to compute means at the end of the study. All statistical tests and values are included in Supplementary Table 2 and Source data file.

## Ethics statement

All animal experiments were approved by the University of Wollongong Animal Ethics Committee (approval number: AEPR21/10) and complied with the Australian National Health and Medical Research Centre code of practice for the care and use of animals for scientific purposes.

## Data availability

The authors declare that all data supporting the findings of this study are available within the paper and its Supplementary Information. Source data are provided with this paper as a Source data file. Source data are provided with this paper.

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

## Acknowledgements

This work was supported by a FightMND Drug Development Grant (DDG-137awarded to J.J.Y., K.V.P., N.R.C. and J.S.L.) and an Australian National Health and Medical Research Council (NHMRC) Investigator Grant (APP1194872 awarded to J.J.Y. and H.E.). J.S.L. was supported by a Motor Neuron Disease Research Institute of Australia Bill Gole Postdoctoral Fellowship (PDF2307). L.M.I. was supported by the Australian Research Council (DP210101957) and Macquarie University. The authors would like to thank Dr Claire Stevens (SMAH Histology) and Dr Nadia Suarez-Bosche for technical assistance and use of microscopy equipment within the Imaging Facility, University of Wollongong. The authors would like to acknowledge the care and technical expertise of the Molecular Horizons Animal Facility Staff. The authors would like to specifically acknowledge Professor Justin Yerbury. This work was the brainchild of Professor Yerbury, who sadly passed away due to ALS during the project. His contribution to the field of ALS research and his personal courage continue to inspire his research team and the wider ALS community.

## Author contributions

N.R.C. provided the panel of antibodies raised against the electrostatic loop of SOD1. C.G.C. generated scFv sequences from these antibody clones. C.G.C., R.B., M.L.B., E.P., N.E.F. and J.G. performed cell-based studies. F.D. generated transgenic mice harbouring the MisfoldUbL transgene. C.G.C., R.B., M.L.B. and J.S.L. performed animal studies and post-mortem analyses. L.M.I., K.V.P., H.E., D.N.S., L.M., J.S.L., and J.J.Y. provided oversight for the project. C.G.C., D.N.S and J.J.Y. conceptualised the overall study. C.G.C. and R.B. drafted the manuscript, and all authors were involved with revisions.

## Competing interests

N.R.C., C.C.G. and J.J.Y. are named inventors on a provisional patent application (PCT/CA2024/050200) filed by the University of British Columbia and the University of Wollongong and assigned to ProMIS Neurosciences, pertaining to sequences of antibody and ubiquitin ligase fusion proteins for misfolded superoxide dismutase 1. N.R.C. is the Chief Scientific Officer of ProMIS Neurosciences. N.R.C. has received consultation compensation from ProMIS and possesses ProMIS stock and stock options. L.M.I. is the Chief Scientific & Medical Officer, founder and shareholder of Celosia Therapeutics. All other authors declare that they have no competing interests with the contents of this article.
