## [Transparent Peer Review file · Nature Communications]

Development of a targeted BioPROTAC degrader selective for misfolded SOD1

Corresponding Author: Dr Jeremy Lum

Version 0:

Reviewer comments:

Reviewer #1

(Remarks to the Author)

The manuscript "Development of a targeted BioPROTAC degrader selective for misfolded SOD1" by Chisholm et al. presented a targeted protein degradation (TPD) strategy to selectively degrade SOD1 protein in amyotrophic lateral sclerosis (ALS). By engineering a bifunctional BioPROTAC composed of a SOD1-specific intrabody and the E3 ubiquitin ligase CHIP, the study demonstrated efficient clearance of misfolded SOD1 variants (including SOD1G93A and SOD1A4V) across multiple cell lines and in the transgenic mouse models. The lead candidate, MisfoldUbL, reduced insoluble SOD1 aggregates, preserved motor neurons, and delayed disease progression in SOD1G93A mice. While I find much of what the authors have done reasonable, the authors need to provide more substantial evidence supporting their conclusion.

Major concerns:

1. While the study demonstrated partial inhibition of SOD1 aggregation by Bafilomycin A (lysosomal pathway) and complete inhibition by MG132 (UPS pathway), the precise mechanisms of co-activation of both degradation systems by BioPROTAC need to be showed. The article has not conducted experiments to map ubiquitination sites, nor visualized the colocalization of SOD1 aggregates with lysosomal and proteasomal markers.
2. An investigation into hormone-mediated regulation is needed to explain the divergent phenotypic outcomes between male and female mice. Testosterone may enhance MisfoldUbL efficacy by upregulating chaperone expression, while estrogen could impair ligase activity through post-translational modifications. RNA-seq analysis of male/female spinal cords at disease onset could identify sex-biased pathways contributing to these differences.
3. The use of the hSYN1 promoter resulted in a 26-fold lower expression of MisfoldUbL in the spinal cord compared to the brain, potentially limiting motor neuron protection. The spinal cord-specific promoters to achieve desirable transgene expression across motor neuron-rich regions may be necessary for the medical translation.
4. The safety of MisfoldUbL expression in the neurons of spinal cord of wild mice should be verified.
5. The effect of MisfoldUbL on the primary neurons with or without SOD1 expression should be detected using single cell sequencing.

Minor concerns:

1. The excessively narrow error bars in Figure 2D may underestimate variability; replacing them with standard error of the mean (SEM) or confidence intervals (CI) would enhance reproducibility.
2. The study omits critical references to recent breakthroughs (findings on DdCBE off-target effects. 2024 Nature, 627(8002):204-211), which is essential for contextualizing tool safety concerns.
3. The authors should provide more details on the distribution and trafficking of MisfoldUbL within the spine.
4. While SH-SY5Y cells are useful, incorporating primary neuronal cultures could provide more physiologically relevant data.
5. Time-coursed data on the levels of BioPROTACs expression post transfection are necessary to ensure they are within a safe range and do not lead to unintended consequences.
6. The authors should provide thorough comparison with other PROTAC strategies in terms of efficiency, specificity, and safety to highlight the potential advantages in discussion section.

Reviewer #2

(Remarks to the Author)

The manuscript by Chisholm and colleagues describes the development of a misfolding-selective degrader of SOD1. The authors compare scFv sequences derived from antibodies that bind to aggregated SOD1. These were fused to the catalytic domains of several E3 ligases to generate a panel of constructs which were tested for activity in degrading fluorescently-tagged mutant SOD1. The most active scFv and ligase were taken forward to examine activity in more detail, and in animal models of protein aggregation. The rationale for using a biological degrader is the ability to specifically remove aggregates, leaving the wildtype version untouched, as wildtype SOD1 has critical functions. The results are encouraging and suggest that SOD1 aggregates can successfully be reduced by this mechanism. I have the following methodological queries and concerns that should be addressed to strengthen the study.

The authors should demonstrate in the screening assays (Fig 2 and 3) the extent of SOD1 degradation by western blot. There is degradation data in later figures, (eg 4) but an early validation of the fluorescence approach would lend the screens more credibility.

Sup Fig 1. It is not valid to make comparisons between lanes that have been separated. These should be run on a single membrane.

Fig legends or blot images should say what was exactly what was probed on western blots, as this information can be critical to interpretation. This info is especially important for the degrader, was this the tag, CHIP or scFv moieties?

Fig 4B convincingly shows the degradation of mutant SOD1, and its dependence on the CHIP ligase. What was the effect on wildtype SOD1? This is critical information for the premise of the approach, that degrading monomers should be avoided.

Sup Fig 1D – was there a transfer issue in the untransfected lane or should this be interpreted that there is more SOD1 expression when BPs are expressed? This should be re-run to verify this result

Reviewer #3

(Remarks to the Author)

The authors have described a novel protein degradation approach to degrading aggregated mutant SOD1 protein from cells. Impressively, their approach appears to aid in the removal of mutant SOD1 protein but has little effect on wild type SOD1, at least in cultured artificial preparations. This approach might be viewed as a therapy that could be “added” to the impressive therapeutic efficacy of the FDA approved toferson, SOD1 antisense therapy. Of course, what's not really known is how important the aggregated SOD1 protein is with regards to actual human cytotoxicity, as clearly eliminating new synthesis of the mutated protein, with the ASO already has impressive therapeutic outcomes. (in fact on true human tissue i.e.autopsy) there is not a lot great evidence for substantial aggregated protein (not the occasional high power picture of a signal neuron!) so one wonders if this invitro and artifical overexpressed biology has marginal relationship to true human disease.

1. Their approach, when applied to actual SOD1 over expressing mice was effective- -- but only very marginally. The therapy showed exceptionally modest efficacy in vivo when studied in the SOD1 mutant mouse. In fact, it is similar to many dozens of therapies in that mouse which went on to fail in actual human trials. To be clear, the efficacy of this approach would appear to be far less than that seen with antisense oligonucleotides. An ideal experiment would be a head-to-head comparison with the previously employed SOD1 antisense- such a study would provide necessary evidence (or not!) on the real utility of this approach. They could do a direct comparison of the treatment of mice with the approved SOD1 antisense oligo compared to their drug- a feasible and valuable experiment. This is comparison would be important as they argue their new therapy can provide benefit in patients, yet the data shown might suggest the clinical efficacy in vivo is really marginal?? As shown in this study-their appears to be little in vivo efficacy compared to antisense oligo nucleotide therapy and makes one suspicious there would be marginal if at all therapeutic value of this in humans.
2. In addition, it would be valuable to test this approach in true human spinal neurons such as SOD1 mutant iPS neurons to determine if efficacy can be seen in authentic human neurons versus all of the overexpression models which are not truly relevant to authentic human disease.
3. Notably, the approach appears to show little effect in their artificial preparations on wild type protein which might be important. It would be important to determine if functional SOD1 activity is altered in the presence of their approach. One might not want to alter wildtype SOD1 function if possible, but at least having that information would be valuable (although, to be clear, the SOD1 ASO, used in patients already, knocks down both wild type and mutant SOD1 production)
4. Is the approach equally effective amongst different SOD1 mutations, for example, at least in human, A5V is a highly toxic mutation while D90A is not—does these approach different is eliminating toxic aggregated mutant species?

Version 1:

Reviewer comments:

Reviewer #1

(Remarks to the Author)

The authors have addressed my most concerns.

Reviewer #2

(Remarks to the Author)

The authors have satisfactorily addressed my concerns.

Reviewer #3

(Remarks to the Author)

The authors response to one query:

.While SH-SY5Y cells are useful, incorporating primary neuronal cultures could provide more physiologically relevant data.

While primary neuronal cultures provide more physiologically relevant data than established cell lines, in the case of therapies targeting SOD-ALS, the SOD1G93A mouse is an established and extensively used model that reproducibly recapitulates the clinical features of the disease. This permits evaluation of therapeutic efficacy in a whole organism context, including providing insights on disease progression that is not possible for primary neuronal cultures. We felt that there was strong evidence in 5our established cell culture data that warranted assessment in an in vivo disease-relevant model.

Actually for eventual human studies the authors are wrong—in fact, as an example, ASO targeting SOD1 were quiet effective in the mutant SOD1 mouse model—but they did not equally wally work in human cells, requiring either a humanized mouse for true target engagement—the authors should in fact test ther drugs in ideally human iPS neuronal cell lines (e.g. neuronal). Its is known in the community that many needed to be tested in vivo and in human cells to eventually identify the most potent candidate.

Their approach remains terribly weak. Perhaps they may optimize their agent=-- or not—they need to be very clear and point out how weak this therapy approach is – as presented in their study (and not some future guess as to better potency) Furthrmore-0 they are wrong about lowering WTS OD1 may it be toxic—the SOD1 knockout mice lives a very long life with only a minor neuropathy in very aged animals. Lowering SOD1, at least sin the CNS by ASO- is simply not a real concern—so the long term concerns they note are not a concern—what remains is that they present in this study a very weak agent. Furthermore, a more potent and long-lasting AAV gene therapy studies are soon to begin (Insemed and Regeneron) by multiple companies for permanent long term knockdown.

REVIEWER COMMENTS

Reviewer #1

1. While the study demonstrated partial inhibition of SOD1 aggregation by Bafilomycin A (lysosomal pathway) and complete inhibition by MG132 (UPS pathway), the precise mechanisms of co-activation of both degradation systems by BioPROTAC need to be showed. The article has not conducted experiments to map ubiquitination sites, nor visualized the colocalization of SOD1 aggregates with lysosomal and proteasomal markers.

The reviewer raises an important point about how SOD1 degradation occurs in the cell and the mechanism by which our BioPROTAC operates. To further investigate our hypothesis that the presence of the BioPROTAC facilitates degradation of mutant SOD1 predominantly by the UPS, we performed the orthogonal approach of immunoblotting lysates from cells treated with MG132 or Bafilomycin. Confirming the results we saw using microscopy, we found that in the presence of proteasome inhibitor but not lysosome inhibitor, the BioPROTAC loses efficacy in reducing mutant soluble SOD1. In contrast, the mutant SOD1 degradation effect of BP2ΔCHIP, without the E3 ligase component, is not altered by the presence of proteasome inhibitor but loses efficacy in cells treated with lysosome inhibitor. The immunoblots have been added to the results with the following statement (page 14 line 251-252): "Similarly, SOD1^{A4V} clearance was abolished when HEK293 cells transfected with BP2 were exposed to MG132 but not when treated with Baf.A1 (Figure 4E).

In the emerging field of BioPROTAC research, the loss of efficacy of potential therapeutics after proteasomal or lysosomal inhibition is routinely used to confirm UPS or autophagy-lysosomal (ALS) degradation, and we direct the reviewer to the following recent studies published in Nature Communications and elsewhere where similar experiments have been used to confirm BioPROTAC mechanism of action:

- Fletcher, A. *et al.* A TRIM21-based bioPROTAC highlights the therapeutic benefit of HuR degradation. *Nat Commun* **14**, 7093 (2023)
- Chan, A. *et al.* Lipid-mediated intracellular delivery of recombinant bioPROTACs for the rapid degradation of undruggable proteins. *Nat Commun* **15**, 5808 (2024).
- Ke, Y *et al.* Targeting 14-3-3θ-mediated TDP-43 pathology in amyotrophic lateral sclerosis and frontotemporal dementia mice. *Neuron* **112**, 1249 (2024).
- Yao, D. *et al.* Selective degradation of hyperphosphorylated tau by proteolysis-targeting chimeras ameliorates cognitive function in Alzheimer's disease model mice. *Frontiers in Pharmacology*, Volume 15 - 2024.

The reviewer's suggestion to map ubiquitination sites on aggregated SOD1 has been attempted by us and others previously without success due to the low abundance of aggregates and the difficulty resolubilising for effective mass spectrometry. However, we have previously published work showing that mutant SOD1 co-localises rapidly with ubiquitin into inclusions and that these inclusions contain both K48 (UPS-related) and K63 (ALP-related) -linked ubiquitin, indicating both systems are involved in the clearance of mutant SOD1. Furthermore, we showed the ubiquitin modified proteome is altered in cells expressing mutant SOD1¹

2. An investigation into hormone-mediated regulation is needed to explain the divergent phenotypic outcomes between male and female mice. Testosterone may enhance MisfoldUbl efficacy by upregulating chaperone expression, while estrogen could impair ligase activity through post-translational modifications. RNA-seq analysis of male/female spinal cords at disease onset could identify sex-biased pathways contributing to these differences.

As with other neurodegenerative diseases including Alzheimer's and Parkinson's disease, sex specific differences are observed in human ALS patients², however the cause of these differences is not yet fully understood³⁻⁶. Sex specific differences are also well documented in the SOD1^{G93A} mouse model^{7,8} and various studies have attempted to elucidate the cause of these differences,⁹⁻¹¹, however no definitive protective or detrimental contribution has been ascertained¹². Several preclinical studies have provided evidence of sex-specific treatment efficacy in SOD1-ALS¹³⁻¹⁶, however the underlying pathophysiology of these differences are currently an understudied area and there remains a paucity of reproducible sex-specific clinical evaluation, leading to calls for sex to be incorporated as an independent variable in all ALS preclinical studies and clinical trial outcomes¹⁷. Despite clear sex-specific evidence in ALS and rodent models, and in addition to being a part of the ALS preclinical and ARRIVE guidelines, many preclinical studies fail to analyse or report these results. For transparency and scientific accuracy, we chose to present our results sex disaggregated – whilst interesting, investigating the mechanisms behind the sex differences are beyond the scope of this study.

We have now added the following statement to the discussion (page 25, lines 412-415): **There is increasing evidence that sex plays a significant role in ALS in both humans and the SOD1^{G93A} mouse model. In this study, we found that while MisfoldUbl conferred protection to both sexes, the impact was greatest for male mice. The physiological reasons underlying sex-specific differences in ALS are yet to be elucidated.**

3. The use of the hSYN1 promoter resulted in a 26-fold lower expression of

MisfoldUbL in the spinal cord compared to the brain, potentially limiting motor neuron protection. The spinal cord-specific promoters to achieve desirable transgene expression across motor neuron-rich regions may be necessary for the medical translation.

As the reviewer points out, in this study, the human synapsin 1 promoter resulted in predominantly cortical expression. While this may be due to neuron specific differences in expression, it is also noted that density of neurons is much lower in the spinal cord than cortex and expression from hsyn at the cellular level is comparable (unpublished data). We have since performed further analysis of the spinal cord in response to the reviewer's concern and determined that although spinal cord expression is lower than in the brain, the highest proportional expression is in the lumbar region (Supplementary Figure 5E), which corresponds to the region of maximal pathological alleviation provided by MisfoldUbL. We are currently testing different promoters which may express more actively in the spinal cord as we continue investigating MisfoldUbL as a potential therapeutic.

4. The safety of MisfoldUbL expression in the neurons of spinal cord of wild mice should be verified.

We have now measured motor neuron count in the ventral horn of the lumbar spinal cord for WT/WT and WT/MisfoldUbL mice and did not find any evidence of a detrimental effect caused by MisfoldUbL expression. The text has also been amended to include the statement (page 20, lines 356-358): "For the WT/MisfoldUbL and WT/WT control groups there was no significant difference in motor neuron number at either the early symptomatic stage or when the mice had reached 200 days old (Supplementary Figure 5F)." Furthermore, there was no difference observed in lifespan or motor function in the WT/MisfoldUbL mice indicating the presence of the MisfoldUbL transgene did not cause toxicity.

5. The effect of MisfoldUbL on the primary neurons with or without SOD1 expression should be detected using single cell sequencing.

It is not clear what insight would be gained from single cell sequencing following MisfoldUbL expression, given the mechanism of action is via post-translational modification of the SOD1 target. Further, we have previously shown significant changes in the Ub-modified proteome in cells expressing mutant forms of SOD1,¹ underscoring the importance of post-translational effects in this context. Single cell transcriptomic studies in an iPSC cell line derived from a patient with the SOD1^{E100G} mutation¹⁸, and from brainstems of SOD1^{G93A} and SOD1^{WT} mice¹⁹ have shown

differentially expressed genes across numerous ALS-associated pathological pathways and processes including synaptic organisation and transmission, cytoskeletal integrity, mitochondrial function and autophagy, highlighting the complex pathogenesis of this disease and underscoring the challenge of relating gene expression to downstream functional consequences including protein interactions, post-translational modifications and phenotypic outcomes.

Minor concerns:

1. The excessively narrow error bars in Figure 2D may underestimate variability; replacing them with standard error of the mean (SEM) or confidence intervals (CI) would enhance reproducibility.

The error bars on Figure 2D, as for all the figures in the manuscript, are standard error of the mean (SEM).

2. The study omits critical references to recent breakthroughs (findings on DdCBE off-target effects. 2024 Nature, 627(8002):204-211), which is essential for contextualizing tool safety concerns.

The reviewer refers to two separate papers in this comment. The 2024 paper in Nature, (627(8002):204-211) is an investigation into a traditional small molecule PROTAC that upon further investigation was found to function as a molecular glue rather than a PROTAC. This "off-target effect" was caused by an incomplete understanding of the interaction between the E3 ligase ligand and the protein being targeted.

In the case of biological PROTACs as is described here, the E3 ligase is fused to the target protein binder, therefore, this strategy permits less uncertainty about off target recruitment. To assess promiscuous activity of the truncated E3 ligase CHIP used in our BioPROTAC we measured Hsp70 levels, a known CHIP substrate and found no effect (Figure 4B) suggesting endogenous substrates of CHIP are not targeted by the BioPROTAC. We also measured WT SOD1 degradation and while we saw small reductions (<8%, Figure 2A and Supplementary Figure 5), as WT SOD1 folds, it transitions through states that may expose our binding epitope, so this was not an unexpected finding. This small reduction of WT SOD1 indicates strong target affinity by the BioPROTAC binding component for misfolded SOD1.

Nevertheless, we have included a statement in our discussion about the safety risk of possible off-target effects (pages 25-26, lines 416-429):

Small-molecule PROTACs have been the subject of intense research over the past two decades with 34 reaching clinical trial in the past 5 years²⁰, although only one of which is for a neurodegenerative disease-associated protein. This highlights the additional challenges faced by targeting proteins in the CNS. Small-molecule PROTACs have several disadvantages in the context of many neurodegenerative disease associated proteins; they rely on a structurally defined, comparatively rigid binding pocket on the target protein, they are susceptible to off-target effects and they require careful investigation to ensure substrate specificity between ligands and E3 ligases/protein of interest²¹. Furthermore, they demonstrate a hook effect where saturating doses of PROTAC reduce degradation efficiency²². In contrast, biological PROTACs comprising of a genetic fusion of POI-binder to truncated E3 ligase, specifically bind the target protein with higher affinity than small molecules, do not require a binding pocket, can recognise a misfolded form of a protein over the WT form, are less demanding to design and synthesize and display superior safety and less toxicity²³. Nevertheless, rigorous validation of specificity and functional consequences is essential in the development of safe and effective biological PROTAC therapies.

We are somewhat perplexed by the query regarding off target effects of DdCBE, a base editing system for mitochondrial DNA and not the subject of the cited study. Given the BioPROTAC system does not target DNA editing nor mitochondria, it is not clear how this system is relevant to the work presented in this manuscript.

3. The authors should provide more details on the distribution and trafficking of MisfoldUbl within the spine.

We have now included additional data addressing this suggestion. Immunoblotting shows that MisfoldUbl is expressed across cervical, thoracic and lumbar regions of the spinal cord We have included a statement in the results (page 20 line 348-349) "expression of the MisfoldUbl was observed across all regions of the spinal cord (Supplementary Figure 5E)."

4. While SH-SY5Y cells are useful, incorporating primary neuronal cultures could provide more physiologically relevant data.

While primary neuronal cultures provide more physiologically relevant data than established cell lines, in the case of therapies targeting SOD-ALS, the SOD1^{G93A} mouse is an established and extensively used model that reproducibly recapitulates the clinical features of the disease. This permits evaluation of therapeutic efficacy in a whole organism context, including providing insights on disease progression that is not possible for primary neuronal cultures. We felt that there was strong evidence in

our established cell culture data that warranted assessment in an *in vivo* disease-relevant model.

5. Time-coursed data on the levels of BioPROTACs expression post transfection are necessary to ensure they are within a safe range and do not lead to unintended consequences.

We have now conducted a time course experiment in cells to assess toxicity. We assessed cell count for untransfected cells and those transfected with BP2 (MisfoldUbl) over both 24 and 48 h and found no difference, see Figure 2I. We have amended the results accordingly (page 8, lines 145-147): "To investigate cell toxicity, we compared cell counts between untransfected and cells transfected with BP2 over 24 and 48 h and found no difference (Figure 2I)." Furthermore, there was no difference observed in lifespan or motor function in the WT/MisfoldUbl mice indicating the presence of the MisfoldUbl transgene was not detrimental or toxic.

6. The authors should provide thorough comparison with other PROTAC strategies in terms of efficiency, specificity, and safety to highlight the potential advantages in discussion section.

We have now included a comparison of various PROTAC strategies in the discussion (as word limits permit, and including references for two excellent reviews on PROTACs and BioPROTACs) as follows (pages 25, lines 416-429)

"Small-molecule PROTACs have been the subject of intense research over the past two decades with 34 reaching clinical trial in the past 5 years (Spitz 2025), although only one of these is for a neurodegenerative disease-associated protein, highlighting the additional challenges faced by targeting proteins in the CNS. Small-molecule PROTACs have several disadvantages in the context of many neurodegenerative disease associated proteins; they rely on a structurally defined, comparatively rigid binding pocket on the target protein, they are susceptible to off-target effects and they require careful investigation to ensure substrate specificity between ligands and E3 ligases/POIs (Hsia et al 2024). Furthermore, they demonstrate a hook effect where saturating doses of PROTAC reduce degradation efficiency (Pettersson et al 2019). In contrast, biological PROTACs comprising of a genetic fusion of POI-binder to truncated E3 ligase, specifically bind the target protein with higher affinity than small molecules, do not require a binding pocket, can recognise a misfolded form of a protein over the WT form, are less demanding to design and synthesize and they display superior safety and less toxicity (reviewed in Wang et al 2023, Li et al 2023). Nevertheless, rigorous validation of specificity and functional consequences is essential in the development of safe and effective biological PROTAC therapies."

Reviewer #2

1. The authors should demonstrate in the screening assays (Fig 2 and 3) the extent of SOD1 degradation by western blot. There is degradation data in later figures, (eg 4) but an early validation of the fluorescence approach would lend the screens more credibility.

We have performed these immunoblots and included them in **Figure 2C and 3G** as recommended.

2. Sup Fig 1. It is not valid to make comparisons between lanes that have been separated. These should be run on a single membrane.

We have now included the full blot in **Supplementary Figure 1** including identification of a treatment that was not carried forward in the study.

3. Fig legends or blot images should say what was exactly what was probed on western blots, as this information can be critical to interpretation. This info is especially important for the degrader, was this the tag, CHIP or scFv moieties?

We have amended all blots to clearly show the antibody that was used as a probe.

4. Fig 4B convincingly shows the degradation of mutant SOD1, and its dependence on the CHIP ligase. What was the effect on wildtype SOD1? This is critical information for the premise of the approach, that degrading monomers should be avoided.

The selectivity for mutant SOD1 and protection of functional WT SOD1 from BP-facilitated proteasomal degradation is a critical aspect of our approach. We have conducted two further experiments investigating the effect of the BP on WT SOD1; protein level (immunoblots) and activity (in gel zymography) (now included in **Supplementary Figure 4**). These experiments confirm that the reduction of WT SOD1 by BP2 is minimal although not absent. This low-level reduction was also evident in our microscopy results and is possibly explained by the exposure of the epitope as WT SOD1 passes through folding stages. Furthermore, the zymography results show that dimeric WT SOD1 is still functionally active when BP2 is present.

5. Sup Fig 1D – was there a transfer issue in the untransfected lane or should this be interpreted that there is more SOD1 expression when BPs are expressed? This should be re-run to verify this result

We have re-run the samples and now display a clearer blot showing expression of BPs has no effect on endogenous SOD1.

Reviewer #3

1. Their approach, when applied to actual SOD1 over expressing mice was effective- -- but only very marginally. The therapy showed exceptionally modest efficacy in vivo when studied in the SOD1 mutant mouse. In fact, it is similar to many dozens of therapies in that mouse which went on to fail in actual human trials. To be clear, the efficacy of this approach would appear to be far less than that seen with antisense oligonucleotides. An ideal experiment would be a head-to-head comparison with the previously employed SOD1 antisense- such a study would provide necessary evidence (or not!) or the real utility of this approach. They could do a direct comparison of the treatment of mice with the approved SOD1 antisense oligo compared to their drug- a feasible and valuable experiment. This is comparison would be important as they argue their new therapy can provide benefit in patients, yet the data shown might suggest the clinical efficacy in vivo is really marginal?? As shown in this study-their appears to be little in vivo efficacy compared to antisense oligo nucleotide therapy and makes one suspicious there would be marginal if at all therapeutic value of this in humans.

The reviewer correctly points out that the effect of our BioPROTAC in the SOD1^{G93A} mouse model was marginal. In this manuscript, we present the BioPROTAC "MisfoldUbl" as a proof-of-concept therapeutic, however, we concur that further optimisation is required. It is important to note that our CRISPR knock-in was a single copy of the BioPROTAC up against 20 copies of mutant SOD1. Taking this into account, we are currently in the early process of selecting a more comprehensively expressed promoter and a therapeutically tractable delivery method to increase biodistribution and expression levels of the BioPROTAC. In line with improvement of our BioPROTAC, the first generation of ASO's targeting SOD1 also showed only modest effects²⁴ before advances in ASO technology identified other, more effective ASO designs.

Although the ASO, Tofersen was recently approved for use in the US and Europe, an important finding to note from the VALOR trial is that the treatment only led to a 20-40% reduction of CSF SOD1, suggesting incomplete silencing. In this scenario, misfolded and aggregated toxic SOD1 species are expected to continually accumulate, contributing to further neuronal degeneration. Simply increasing the ASO dosage to achieve greater reduction in SOD1 content may result in collateral deleterious effects by unscavenged superoxide free radicals e.g., the iSODDES syndrome²⁵. For many neurodegenerative disease-associated proteins, removing the functioning WT form is problematic, hence the interest in biological PROTACs and their ability to specifically reduce the levels of only the misfolded form of proteins.

In the current stage of development for the MisfoldUbL, we believe it is not feasible to make comparisons to other therapies due to our resources being focused on improving the system. Furthermore, Tofersen is currently under patent and difficult to access for suggested studies as there is currently only one published SOD1 mouse study that has investigated Tofersen²⁴. At this point we believe it is not warranted to perform a 200-day head-to-head comparison study between the MisfoldUbL and Tofersen. While we hold much personal hope that all mutant SOD1 cases of ALS will be treatable with Tofersen, we strongly believe that it is imperative that we continue to examine alternatives with distinct mechanisms of action that could also be used in combination with or as an improvement to Tofersen.

2. In addition, it would be valuable to test this approach in true human spinal neurons such as SOD1 mutant iPSC neurons to determine if efficacy can be seen in authentic human neurons versus all of the overexpression models which are not truly relevant to authentic human disease.

In this proof-of-concept study, we describe a BioPROTAC therapeutic that engages misfolded SOD1 selectively, spares WT SOD1, and produces measurable functional benefit *in vivo* — key translational milestones that address many of the limitations associated with using iPSC models alone. However, we agree that assessing this therapeutic in iPSC-derived motor neurons from SOD1-ALS patients would be productive, particularly to validate compatibility with human cellular machinery. We are currently pursuing this strategy in our further investigations and optimisations of the MisfoldUbL.

3. Notably, the approach appears to show little effect in their artificial preparations on wild type protein which might be important. It would be important to determine if functional SOD1 activity is altered in the presence of their approach. One might not want to alter wildtype SOD1 function if possible, but at least having that information would be valuable (although, to be clear, the SOD1 ASO, used in patients already, knocks down both wild type and mutant SOD1 production)

The selectivity for mutant SOD1 and protection of functional WT SOD1 from BP-facilitated proteasomal degradation is a critical aspect of our approach. In addition to experiments showing preservation of endogenous SOD1 in the presence of the BioPROTACs (Supplementary Figure 1), and the minimal reduction of SOD1^{WT} in fluorescence microscopy assays (Figure 2a), we have now conducted two further experiments investigating the effect of the BP on WT SOD1; immunoblots and in gel zymography (Supplementary Figure 4). These experiments confirm that the reduction

of WT SOD1 by BP2 is minimal although not absent. The zymography results show that dimeric WT SOD1 is still functionally active when BP2 is present.

4. Is the approach equally effective amongst different SOD1 mutations, for example, at least in human, A5V is a highly toxic mutation while D90A is not—does these approach different is eliminating toxic aggregated mutant species?

We refer the reviewer to Figure 1J. BP2 was found to be effective at reducing the number of insoluble aggregates across a range of nine different SOD1 mutants ranging from 40-60%.

References

- 1 Farrawell, N. E. *et al.* SOD1(A4V) aggregation alters ubiquitin homeostasis in a cell model of ALS. *J Cell Sci* **131** (2018). <https://doi.org/10.1242/jcs.209122>
- 2 Longinetti, E. & Fang, F. Epidemiology of amyotrophic lateral sclerosis: an update of recent literature. *Curr Opin Neurol* **32**, 771-776 (2019).
<https://doi.org/10.1097/wco.0000000000000730>
- 3 Garton, F. C., Trabjerg, B. B., Wray, N. R. & Agerbo, E. Cardiovascular disease, psychiatric diagnosis and sex differences in the multistep hypothesis of amyotrophic lateral sclerosis. *Eur J Neurol* **28**, 421-429 (2021). <https://doi.org/10.1111/ene.14554>
- 4 Grad, L. I., Rouleau, G. A., Ravits, J. & Cashman, N. R. Clinical Spectrum of Amyotrophic Lateral Sclerosis (ALS). *Cold Spring Harb Perspect Med* **7** (2017).
<https://doi.org/10.1101/cshperspect.a024117>
- 5 Grassano, M. *et al.* Sex Differences in Amyotrophic Lateral Sclerosis Survival and Progression: A Multidimensional Analysis. *Ann Neurol* **96**, 159-169 (2024).
<https://doi.org/10.1002/ana.26933>
- 6 Ortholand, J., Pradat, P. F., Tezenas du Montcel, S. & Durrleman, S. Interaction of sex and onset site on the disease trajectory of amyotrophic lateral sclerosis. *J Neurol* **270**, 5903-5912 (2023). <https://doi.org/10.1007/s00415-023-11932-7>
- 7 Weydt, P., Hong, S. Y., Kliot, M. & Möller, T. Assessing disease onset and progression in the SOD1 mouse model of ALS. *Neuroreport* **14**, 1051-1054 (2003).
<https://doi.org/10.1097/01.wnr.0000073685.00308.89>
- 8 Pfohl, S. R., Halicek, M. T. & Mitchell, C. S. Characterization of the Contribution of Genetic Background and Gender to Disease Progression in the SOD1 G93A Mouse Model of Amyotrophic Lateral Sclerosis: A Meta-Analysis. *J Neuromuscul Dis* **2**, 137-150 (2015).
<https://doi.org/10.3233/jnd-140068>
- 9 Tomas, D. *et al.* Dissociation of disease onset, progression and sex differences from androgen receptor levels in a mouse model of amyotrophic lateral sclerosis. *Sci Rep* **11**, 9255 (2021). <https://doi.org/10.1038/s41598-021-88415-0>
- 10 McLeod, V. M. *et al.* Androgen receptor antagonism accelerates disease onset in the SOD1(G93A) mouse model of amyotrophic lateral sclerosis. *Br J Pharmacol* **176**, 2111-2130 (2019). <https://doi.org/10.1111/bph.14657>
- 11 Yoo, Y. E. & Ko, C. P. Dihydrotestosterone ameliorates degeneration in muscle, axons and motoneurons and improves motor function in amyotrophic lateral sclerosis model mice. *PLoS One* **7**, e37258 (2012). <https://doi.org/10.1371/journal.pone.0037258>
- 12 Vegeto, E. *et al.* The Role of Sex and Sex Hormones in Neurodegenerative Diseases. *Endocr Rev* **41**, 273-319 (2020). <https://doi.org/10.1210/endrev/bnz005>
- 13 Bame, M., Pentiak, P. A., Needleman, R. & Brusilow, W. S. Effect of sex on lifespan, disease progression, and the response to methionine sulfoximine in the SOD1 G93A mouse model for ALS. *Genet Med* **9**, 524-535 (2012). <https://doi.org/10.1016/j.genm.2012.10.014>
- 14 Bartlett, R., Sluyter, V., Watson, D., Sluyter, R. & Yerbury, J. J. P2X7 antagonism using Brilliant Blue G reduces body weight loss and prolongs survival in female SOD1(G93A) amyotrophic lateral sclerosis mice. *PeerJ* **5**, e3064 (2017). <https://doi.org/10.7717/peerj.3064>
- 15 Butovsky, O. *et al.* Targeting miR-155 restores abnormal microglia and attenuates disease in SOD1 mice. *Ann Neurol* **77**, 75-99 (2015). <https://doi.org/10.1002/ana.24304>
- 16 Ruiz-Ruiz, C. *et al.* Chronic administration of P2X7 receptor antagonist JNJ-47965567 delays disease onset and progression, and improves motor performance in ALS SOD1(G93A) female mice. *Dis Model Mech* **13** (2020). <https://doi.org/10.1242/dmm.045732>
- 17 Zamani, A., Thomas, E. & Wright, D. K. Sex biology in amyotrophic lateral sclerosis. *Ageing Res Rev* **95**, 102228 (2024). <https://doi.org/10.1016/j.arr.2024.102228>

- 18 Namboori, S. C. *et al.* Single-cell transcriptomics identifies master regulators of neurodegeneration in SOD1 ALS iPSC-derived motor neurons. *Stem Cell Reports* **16**, 3020-3035 (2021). <https://doi.org/10.1016/j.stemcr.2021.10.010>
- 19 Liu, W. *et al.* Single-cell RNA-seq analysis of the brainstem of mutant SOD1 mice reveals perturbed cell types and pathways of amyotrophic lateral sclerosis. *Neurobiology of Disease* **141**, 104877 (2020). <https://doi.org/https://doi.org/10.1016/j.nbd.2020.104877>
- 20 Spitz, M. L., Aseel, K. & and Benhamou, R. I. Advancing target validation with PROTAC technology. *Expert Opinion on Drug Discovery* **20**, 551-563 (2025). <https://doi.org/10.1080/17460441.2025.2490248>
- 21 Hsia, O. *et al.* Targeted protein degradation via intramolecular bivalent glues. *Nature* **627**, 204-211 (2024). <https://doi.org/10.1038/s41586-024-07089-6>
- 22 Pettersson, M. & Crews, C. M. PROteolysis TArgeting Chimeras (PROTACs) - Past, present and future. *Drug Discov Today Technol* **31**, 15-27 (2019). <https://doi.org/10.1016/j.ddtec.2019.01.002>
- 23 Wang, H. *et al.* Beyond canonical PROTAC: biological targeted protein degradation (bioTPD). *Biomaterials Research* **27**, 72 (2023). <https://doi.org/10.1186/s40824-023-00385-8>
- 24 McCampbell, A. *et al.* Antisense oligonucleotides extend survival and reverse decrement in muscle response in ALS models. *J Clin Invest* **128**, 3558-3567 (2018). <https://doi.org/10.1172/jci99081>
- 25 Park, J. H. *et al.* The motor system is exceptionally vulnerable to absence of the ubiquitously expressed superoxide dismutase-1. *Brain Commun* **5**, fcad017 (2023). <https://doi.org/10.1093/braincomms/fcad017>

NCOMMS-25-08111A: Authors Response to Reviewers

Reviewer #1 (Remarks to the Author)

The authors have addressed my most concerns.

Reviewer #2 (Remarks to the Author)

The authors have satisfactorily addressed my concerns.

Reviewer #3 (Remarks to Author)

The authors response to one query:

While SH-SY5Y cells are useful, incorporating primary neuronal cultures could provide more physiologically relevant data. While primary neuronal cultures provide more physiologically relevant data than established cell lines, in the case of therapies targeting SOD-ALS, the SOD1G93A mouse is an established and extensively used model that reproducibly recapitulates the clinical features of the disease. This permits evaluation of therapeutic efficacy in a whole organism context, including providing insights on disease progression that is not possible for primary neuronal cultures. We felt that there was strong evidence in our established cell culture data that warranted assessment in an in vivo disease-relevant model.

Actually for eventual human studies the authors are wrong—in fact, as an example, ASO targeting SOD1 were quite effective in the mutant SOD1 mouse model—but they did not equally well work in human cells, requiring either a humanized mouse for true target engagement—the authors should in fact test their drugs in ideally human iPSC neuronal cell lines (e.g. neuronal). It is known in the community that many needed to be tested in vivo and in human cells to eventually identify the most potent candidate.

The first generation SOD1-ASO was tested for SOD1 reduction in mice, rats, primates and human fibroblasts from a single patient with the SOD1A4V mutation¹, before being tested for safety in a phase 1 clinical trial in humans². The authors noted that the effects in the animal models were modest prompting the development of second generation ASOs. These were screened in SH-SY5Y cells, with the most effective (tofersen), then tested for efficacy in mice, rats and primates³. Tofersen was not tested in iPSCs before human clinical trials.

We agree that iPSC's are a useful resource for screening possible therapies and have included a statement to this effect in our manuscript (Discussion, Paragraph 3). However, SOD1 iPSCs have a much more subtle phenotype compared to overexpression models in established cell lines and do not fully recapitulate SOD1 pathology⁴. Given the range of SOD1 mutations causative for SOD1-ALS, we wanted to assess the efficacy of our BioPROTAC against as many SOD1 variants as possible, an approach that would be expensive and time consuming in iPSCs. Furthermore, the overexpression model used in transient transfection systems is ideal for challenging a SOD1 targeting therapeutic such as our BioPROTAC as it produces robust results across a range of meaningful phenotypes (SOD1 levels, aggregation, target engagement).

In line with this, whilst we acknowledge that the SOD1G93A mouse has its limitations, it remains a powerful model for testing SOD1-targeting therapies. In our work, we incorporated a single copy of

our BioPROTAC in mice that harbour over 20 copies of mutant SOD1. Despite this, our results show target engagement, delayed disease onset, improved motor function, and a reduction in mutant SOD1 levels in these mice, therefore providing strong support for our proof-of-concept BioPROTAC. While iPSCs are useful for screening potential ALS targeting therapies and assessing effect in a humanised model, the SOD1G93A mouse model remains critical for evaluating therapeutic efficacy and disease progression in a whole-organism setting. Our rationale to progress directly from reproducible SH-SY5Y cell culture data to *in vivo* evaluation in this established disease model is scientifically sound, consistent with best practice in the field and appropriate for the scope of this study.

Their approach remains terribly weak. Perhaps they may optimize their agent— or not—they need to be very clear and point out how weak this therapy approach is – as presented in their study (and not some future guess as to better potency)

We respectfully disagree with the reviewer’s comment that our approach is “terribly weak”. Our data demonstrate clear, reproducible target engagement and therapeutic effect in both cell-based assays and *in vivo* models, providing strong evidence that the strategy is viable.

As with any novel therapeutic approach, further optimisation is expected and necessary, however the results presented here establish proof-of-concept and strongly support continued development. We have been transparent in acknowledging the limitations of our study, including the lack of effect of the BioPROTAC on mice survival (which we note was due to weight-loss and not motor impairment), our scFv targeting the electrostatic loop only, and our choice of promoter in mice. Importantly, it is standard practice in ALS therapeutic development for early studies to demonstrate feasibility and biological effect before iterative rounds of optimisation. Our study provides this critical first step. BioPROTAC therapeutics are extremely new to the field of neurodegenerative disease, with only two published studies^{5,6}, we believe it is therefore worthwhile to share this proof-of-concept design. To label the approach as “weak” is inaccurate and risks overlooking the value of establishing a new therapeutic modality for ALS, particularly given the urgent unmet need in this disease.

Furthermore- they are wrong about lowering WTS OD1 may it be toxic—the SOD1 knockout mice lives a very long life with only a minor neuropathy in very aged animals. Lowering SOD1, at least in the CNS by ASO- is simply not a real concern—so the long term concerns they note are not a concern—what remains is that they present in this study a very weak agent. Furthermore, a more potent and long-lasting AAV gene therapy studies are soon to begin (Insemed and Regeneron) by multiple companies for permanent long term knockdown.

Curiously, the reviewer discounts the value of mouse data to assess the potential of a SOD1 targeting therapeutic, but relies solely on mouse data to support the claim that long term knockdown of SOD1 is not a concern. SOD1 knockout mice show a phenotype of progressive muscle atrophy and weakness with preferential denervation of fast-twitch muscles⁷. Furthermore, there are now three peer-reviewed case reports that show homozygous loss of SOD1 in children results in severe motor function loss along with other life-affecting pathologies⁸⁻¹⁰. These studies do raise serious concerns about therapies that rely on lowering levels of active SOD1 in the CNS and indicate that this should be carefully monitored in ongoing treatment using gene silencing strategies. Indeed, this cautious approach is promoted by the designers of Tofersen when they state that “Antisense oligonucleotide infusion can be regulated or stopped should there be any unforeseen side effects, a key consideration for human application”¹.

References:

- 1 Smith, R. A. et al. Antisense oligonucleotide therapy for neurodegenerative disease. *J Clin Invest* 116, 2290-2296 (2006). <https://doi.org/10.1172/jci25424>
- 2 Miller, T. M. et al. An antisense oligonucleotide against SOD1 delivered intrathecally for patients with SOD1 familial amyotrophic lateral sclerosis: a phase 1, randomised, first-in-man study. *Lancet Neurol* 12, 435-442 (2013). [https://doi.org/10.1016/s1474-4422\(13\)70061-9](https://doi.org/10.1016/s1474-4422(13)70061-9)
- 3 McCampbell, A. et al. Antisense oligonucleotides extend survival and reverse decrement in muscle response in ALS models. *J Clin Invest* 128, 3558-3567 (2018). <https://doi.org/10.1172/jci99081>
- 4 Dimos, J. T. et al. Induced pluripotent stem cells generated from patients with ALS can be differentiated into motor neurons. *Science* 321, 1218-1221 (2008). <https://doi.org/10.1126/science.1158799>
- 5 Ke, Y. D. et al. Targeting 14-3-3 θ -mediated TDP-43 pathology in amyotrophic lateral sclerosis and frontotemporal dementia mice. *Neuron* 112, 1249-1264.e1248 (2024). <https://doi.org/10.1016/j.neuron.2024.01.022>
- 6 Jiang, Y. et al. Single-domain antibody-based protein degrader for synucleinopathies. *Molecular Neurodegeneration* 19, 44 (2024). <https://doi.org/10.1186/s13024-024-00730-y>
- 7 Fischer, L. R., Li, Y., Asress, S. A., Jones, D. P. & Glass, J. D. Absence of SOD1 leads to oxidative stress in peripheral nerve and causes a progressive distal motor axonopathy. *Exp Neurol* 233, 163-171 (2012). <https://doi.org/10.1016/j.expneurol.2011.09.020>
- 8 Ezer, S. et al. Infantile SOD1 deficiency syndrome caused by a homozygous SOD1 variant with absence of enzyme activity. *Brain* 145, 872-878 (2022). <https://doi.org/10.1093/brain/awab416>
- 9 Park, J. H. et al. SOD1 deficiency: a novel syndrome distinct from amyotrophic lateral sclerosis. *Brain* 142, 2230-2237 (2019). <https://doi.org/10.1093/brain/awz182>
- 10 Andersen, P. M. et al. Phenotype in an Infant with SOD1 Homozygous Truncating Mutation. *N Engl J Med* 381, 486-488 (2019). <https://doi.org/10.1056/NEJMc1905039>